# ON CORESET FOR LASSO REGRESSION PROBLEM WITH SENSITIVITY SAMPLING

**Yuanbin Zou**
School of Computer Science and Engineering
Central South University
Changsha, China
yuanbinzou@csu.edu.cn

**Junyu Huang**
School of Computer Science and Engineering
Central South University
Changsha, China
junyuhuangcsu@foxmail.com

**Jianxin Wang**
School of Computer Science and Engineering
Central South University
Hunan Provincial Key Lab on Bioinformatics
Xiangjiang Laboratory
Changsha, China
jxwang@mail.csu.edu.cn

**Qilong Feng**[*]
School of Computer Science and Engineering
Central South University
Changsha, China
csufeng@mail.csu.edu.cn

## ABSTRACT

In this paper, we study coreset construction for LASSO regression, where a coreset is a small, weighted subset of the data that approximates the original problem with provable guarantees. For unregularized regression problems, sensitivity sampling is a successful and widely applied technique for constructing coresets. However, extending these methods to LASSO typically requires coreset size to scale with $O(\mathcal{G}d)$, where $d$ is the VC dimension and $\mathcal{G}$ is the total sensitivity, following existing generalization bounds. A key challenge in improving upon this general bound lies in the difficulty of capturing the sparse and localized structure of the function space induced by the $\ell_1$ penalty in LASSO objective. To address this, we first provide an empirical process-based method of sensitivity sampling for LASSO, localizing the procedure by decomposing the functional space into separate components, which leads to tighter estimation error. By carefully leveraging the geometric properties of these localized spaces, we establish tight empirical process bounds on the required coreset size. These techniques enable us to achieve a coreset of size $\tilde{O}(\epsilon^{-2}d \cdot (\log^3 d \cdot \min\{1, \log d/\lambda^2\} + \log(1/\delta)))$, which ensures a $(1 \pm \epsilon)$-approximation for any $\epsilon, \delta \in (0, 1)$ and $\lambda > 0$. Furthermore, we give a lower bound showing that any algorithm achieving a $(1 + \epsilon)$-approximation must select at least $\Omega(\frac{d \log d}{\epsilon^2})$ rows in the regime where $\lambda = O(d^{-1/2})$. Empirical experiments show that our proposed algorithm is at least 4 times faster than the existing LASSO solver and more than 9 times faster on half of the datasets, while ensuring high solution quality and sparsity.

## 1 INTRODUCTION

In machine learning and regression analysis, sparse models have been extensively studied over the past decades. These models typically address issues such as sparse regression (Natarajan, 1995), variable selection (Zou & Hastie, 2005), and multicollinearity (Altelbany, 2021), aiming to improve model interpretability and computational efficiency by reducing the number of features. One of the most widely used methods for solving sparse models is the Least Absolute Shrinkage and Selection Operator (LASSO), which is first introduced in (Tibshirani, 1997). The core idea of LASSO regression is to apply an $\ell_1$-norm penalty, ensuring sparsity by shrinking some coefficients to zero. Therefore, in practice, LASSO is widely applied in sparse models due to its effectiveness in enabling

---

[*]Corresponding Author.

both variable selection and regularization with improved interpretability and prevented overfitting issues. The formal definition of the LASSO regression is given as follows.

**LASSO Regression Problem.** Given an $n \times d$ matrix $A$, an $n$-dimension vector $b$, and a regularization parameter $\lambda > 0$, the goal of LASSO problem is to find a $d$-dimension vector $x$ that minimizes $\|Ax - b\|_2^2 + \lambda \|x\|_1$, where $\|Ax - b\|_2^2$ is the residual sum of squares, and the $\|x\|_1$ denotes the sum of the absolute values of the entries in $x$.

Although LASSO regression has been extensively studied over the past decade, the efficiency of LASSO algorithms for handling large-scale data still depends heavily on the number of rows $n$ of the input matrix $A$. Specifically, the running time of existing algorithms, such as coordinate descent (Friedman et al., 2010), ISTA (Daubechies et al., 2004), and FISTA (Beck & Teboulle, 2009), is typically $O(nT \cdot \text{poly}(d))$, where $T$ denotes the number of iterations. However, for datasets with a large number of samples $n$, LASSO may suffer from scalability issues. Consequently, developing row-subsampling methods for LASSO regression is crucial for improving computational efficiency.

Among the vast literature on large-scale regression tasks, coreset techniques have played a major role in data subsampling. These algorithms aim to construct a weighted subset of rows from both $A$ and $b$, forming a compact representation that effectively approximates the original regression problem with strong theoretical guarantees. Along this line of research, several coreset construction algorithms have been proposed for the $\ell_p$ linear regression (Clarkson, 2005; Drineas et al., 2006; Dasgupta et al., 2009; Cohen & Peng, 2015; Woodruff & Yasuda, 2024; 2023; Munteanu & Omlor, 2024). For regularized regression, Avron et al. (2017) constructed a coreset of size $O(\frac{sd_\lambda(A) + \log(1/\epsilon)}{\epsilon} \log \frac{sd_\lambda(A)}{\epsilon})$ for ridge regression, where $sd_\lambda(A) \leq d$ denotes the statistical dimension of the matrix $A$. Moreover, Kacham & Woodruff (2020) introduced deterministic algorithms for coreset construction and also explored a streaming model for this problem. Curtin et al. (2019) provided a coreset for logistic regression of size $O(d\sqrt{n})$. Chhaya et al. (2020) proposed a coreset based on sensitivity sampling for the norm-based regularized regression problem $\|Ax - b\|_p^p + \lambda \|x\|_p^p$ with $p \geq 2$. In a related recent work, Chhaya et al. (2020) studied a modified LASSO problem by constructing a coreset for the objective $\|Ax - b\|_2^2 + \lambda \|x\|_1^2$. However, the regularization term $\lambda \|x\|_1^2 = \lambda (\sum_i |x_i|)^2$, due to its quadratic form, introduces cross terms among the coordinates $x_i$. This may lead to solutions with substantially more nonzero coefficients than those of standard LASSO, thereby preventing it from promoting sparsity in the same way as the $\ell_1$ norm and weakening its sparsity-inducing effect. To the best of our knowledge, there are currently no relevant theoretical results on coreset construction for standard LASSO, which motivates our work on developing such a coreset.

**Coreset for LASSO.** Let $A \in \mathbb{R}^{n \times d}$ be a matrix and $b \in \mathbb{R}^n$. Define $S \in \mathbb{R}^{n \times n}$ as a diagonal matrix, where the $i$-th row of both $A$ and $b$ are independently sampled with probability $p_i$ for each $i \in [n]$. Let $m$ denote the number of sampled rows. If row $i$ is selected, set $S_{i,i} = 1/\sqrt{mp_i}$, and set $S_{i,i} = 0$ otherwise. We say that $S$ defines an $(\epsilon, \delta)$-coreset for LASSO problem if, with probability at least $1 - \delta$, for all $x \in \mathbb{R}^d$ and $\lambda > 0$, the following holds

$$\|S(Ax - b)\|_2^2 + \lambda \|x\|_1 \in (1 \pm \epsilon) \left( \|Ax - b\|_2^2 + \lambda \|x\|_1 \right),$$

where $\epsilon \in (0, 1)$. The coreset size is defined as the number of nonzero entries on the diagonal of $S$, i.e., the number of sampled rows $m$.

Sensitivity sampling (Feldman & Langberg, 2011; Chhaya et al., 2020; Woodruff & Yasuda, 2023) has been extensively studied in regression without regularization, where rows are sampled in proportion to their importance in the regression objective. A common challenge in directly applying sensitivity sampling to LASSO lies in bounding the generalization error under the $\ell_1$-regularized objective using standard empirical process tools. In the general framework of sensitivity sampling, Braverman et al. (2016) showed that, given sensitivity scores $\{\varrho_i\}_{i=1}^n$, a $(1 \pm \epsilon)$-approximate coreset typically requires a size of $\tilde{O}\left(\frac{\mathcal{G}d}{\epsilon^2}\right)$ [1], where $\mathcal{G}$ is the sum of the sensitivity scores and $d$ denotes the VC dimension of the given problem. This bound arises from applying a union bound to worst-case $\epsilon$-net methods and variance analysis. Consequently, directly applying traditional analysis to LASSO leads to large coreset sizes, which can limit scalability in high-dimensional settings. To address this issue, empirical process techniques and chaining methods have been developed to reduce the dependence on the $\mathcal{G}d$ term in the coreset-size bound (Cohen et al., 2015; Woodruff & Yasuda, 2023; Munteanu & Omlor, 2024; Bansal et al., 2024). However, integrating empirical process theory with

---

[1]We write $\tilde{O}(f(n))$ to denote $O(f(n) \cdot \text{poly} \log f(n))$.

LASSO regression requires addressing the sparse and localized structure of the parameter space induced by the $\ell_1$-penalty. In particular, the functional space $\Omega = \{x \in \mathbb{R}^d \mid h(x) + p(x) \leq R\}$, defined for a fixed radius $R > 0$, is determined by the residual term $h(x) = \|Ax - b\|_2^2$ and the penalty term $p(x) = \lambda\|x\|_1$ in the objective function. The interaction between the residual and penalty terms results in a highly complex geometry for $\Omega$, complicating the standard empirical process analysis. Additionally, the non-smooth boundary induced by the $\ell_1$-penalty leads to large error bounds when applying the Bernstein's inequality and $\epsilon$-net arguments, in (Chhaya et al., 2020). Therefore, developing a sensitivity sampling method for LASSO that constructs a coreset of size smaller than $\tilde{O}(\mathcal{G}d)$ remains a key challenge.

## 1.1 OUR CONTRIBUTION

In this paper, we aim to improve upon existing standard bounds for LASSO coresets, which often yield large coreset sizes due to the use of union-bound-based $\epsilon$-net methods. The main difficulty arises from the intricate structure of the function space induced by both the residual error and the $\ell_1$ regularization term. This complexity makes it difficult to directly apply standard empirical process techniques to sensitivity sampling. To address this issue, we propose a localized empirical process method that reformulates the sensitivity scores and sampling error in a more tractable manner. Specifically, we define a weighted Gaussian-based empirical process for the coreset loss and decompose the overall function space into two separable components: the residual space and the $\ell_1$ penalty space. Each of these components has lower complexity than the original space $\Omega$, allowing for tighter bounds on the Gaussian diameter and metric entropy of each component. By carefully applying symmetrization techniques and leveraging the geometric properties of these localized spaces, we derive upper bounds on the localized Gaussian diameter and metric entropy. With probability at least $1 - \delta$, these bounds allow us to control the sampling error and construct a coreset of size $\tilde{O}(\epsilon^{-2}d \cdot (\log^3 d \cdot \min\{1, \log d/\lambda^2\} + \log(1/\delta)))$, achieving a $(1 \pm \epsilon)$-approximation for all $\epsilon, \delta \in (0, 1)$ and $\lambda > 0$.

To complement our upper bound analysis, we establish a near-matching lower bound on the coreset size for LASSO regression via an information-theoretic method. By reducing the problem to a classical sparse recovery setting, we show that any estimator achieving a $(1 + \epsilon)$-approximation from the coreset must access a minimum number of rows to achieve the sparse recovery task. In particular, in the regime where $\lambda = O(\frac{1}{\sqrt{d}})$, corresponding to the case in which the number of nonzero entries may be large, we prove that the number of required rows is at least $\Omega(\frac{d}{\epsilon^2}\log(d))$. Our coreset size matches this lower bound up to polylogarithmic factors in the dimension $d$. Empirical experiments show that our proposed algorithm is at least 4 times faster than solving LASSO directly on the full dataset and more than 9 times faster on half of the datasets, while preserving high solution quality. Notably, on a dataset with 8 million samples, our method completes in only 15 minutes.

## 1.2 OTHER RELATED WORK

LASSO regression, first introduced in (Tibshirani, 1996), has been widely studied for learning sparse models, such as variable selection (Tibshirani, 1997; Hans, 2010) and compressed sensing (Angelosante et al., 2009). Many optimization algorithms have been developed for LASSO, including the fast iterative shrinkage-thresholding algorithm (Beck & Teboulle, 2009), coordinate descent (Friedman et al., 2010), the smooth $\ell_1$ algorithm (Schmidt et al., 2007), and the path-following algorithm (Tibshirani & Taylor, 2011). LASSO regression uses $\ell_1$-regularization to relax the sparsity penalty (typically denoted by $\|x\|_0$), which is NP-hard (Natarajan, 1995). However, tuning the regularization parameter often leads to high computational costs. To address this, several methods have been proposed. Friedman et al. (2010) provided the `glmnet` package based on coordinate descent for solving LASSO. Obozinski & Bach (2012) proposed a stochastic variant that improves convergence via random selection. Wang et al. (2025) accelerated hyperparameter tuning using Markov resampling. To the best of our knowledge, there is currently no coreset construction method for the standard LASSO problem.

Sensitivity sampling is a well-studied technique for coreset construction in both theory and practice. It was first introduced by Agarwal et al. (2004), and has since been widely applied to various problems, including clustering (Feldman & Langberg, 2011; Braverman et al., 2022; Bansal et al., 2024), linear regression (Drineas et al., 2006; Woodruff & Yasuda, 2024; 2023; Munteanu & Omlor, 2024),

and matrix approximation (Dasgupta et al., 2009; Cohen et al., 2015). For ordinary least-squares regression, (Drineas et al., 2006) proposed a coreset algorithm based on the well-known statistical leverage score sampling. Dasgupta et al. (2009) extended this line of work to $\ell_p$ linear regression using the well-conditioned basis method. More recently, a tight framework for constructing coresets for unregularized regression has been developed in a series of works (Woodruff & Yasuda, 2023; 2024; Munteanu & Omlor, 2024), leveraging chaining techniques from empirical process theory.

Sensitivity sampling techniques have been extensively studied for regularized regression problems. For logistic regression, sensitivity-based sampling has been successfully applied in a series of works Munteanu et al. (2018); Curtin et al. (2019); Mai et al. (2021); Munteanu & Omlor (2024). In particular, Munteanu & Omlor (2024) recently provided a strong coreset of size $\tilde{O}(\mu d/\epsilon^2)$ based on Lewis-weight sampling Parulekar et al. (2021), where $\mu$ captures the complexity of the input data distribution. For ridge regression, Avron et al. (2017) pioneered the use of coreset techniques by showing that a weak coreset of size $\tilde{O}(sd_\lambda(A)/\epsilon^2)$ suffices to achieve a $(1 + \epsilon)$-approximation. Kacham & Woodruff (2020) developed optimal deterministic coreset constructions for multi-response ridge regression. Their method selects $O(sd_\lambda(A)/\epsilon)$ rows and achieves a $(1 + \epsilon)$-approximation, with matching lower bounds that establish the tightness of the dependence on $sd_\lambda(A)$. In the regime where $n \gg d$, the statistical dimension $sd_\lambda(A)$ satisfies $sd_\lambda(A) \leq \text{rank}(A) \leq d$, and larger values of the regularization parameter $\lambda$ can lead to smaller coreset sizes for ridge regression.

In the broader context of norm-regularized regression, Chhaya et al. (2020) considered coreset constructions for $\ell_p$ regularized regression problems of the form $\|Ax - b\|_p^p + \lambda\|x\|_p^p$, where $p > 2$. They provided a coreset of $\tilde{O}(\frac{d^{p+1}}{\epsilon^2 \cdot (1+\lambda/\|A'\|_{(p)}^p)})$ based on sensitivity sampling techniques. Moreover, they first showed that when $r \neq s$, no strong coreset can be smaller than the optimal coreset size for the unregularized term $\|Ax - b\|_p^r$. This result applies in particular to LASSO, where $p = r = 2$ and $q = s = 1$. To address the LASSO objective, Chhaya et al. (2020) proposed a modified formulation in which the regularization term $\|x\|_1$ is replaced by $\|x\|_1^2$, enabling the use of ridge regression coreset techniques Avron et al. (2017) to construct a coreset of size $\tilde{O}(sd_\lambda(A)/\epsilon^2)$. However, this modification introduces cross terms among the components of $x$, which may weaken the sparsity-inducing effect of the standard $\ell_1$ regularization. In this paper, we propose a coreset for the standard LASSO objective with size $\tilde{O}(\epsilon^{-2}d \cdot (\log^3 d \cdot \min\{1, \log d/\lambda^2\} + \log(1/\delta)))$, which preserves the $\tilde{O}(d/\epsilon^2)$ bound when $\lambda$ approaches 0 or $\infty$. In addition, sketching-based methods using randomized projections have also been applied to LASSO in recent work Mai et al. (2023). Designing sensitivity sampling methods for constructing coresets for LASSO remains an interesting open problem.

## 2 PRELIMINARIES

Given a positive integer $n$, let $[n] = \{1, 2, \dots, n\}$. For a $d$-dimensional vector $x \in \mathbb{R}^d$, the $\ell_p$-norm of $x$ is $\|x\|_p = (\sum_{i=1}^d |x_i|^p)^{1/p}$. For an $n \times d$ matrix $A$, the induced $p$-norm is $\|A\|_{(p)}$, which is defined as $\|A\|_{(p)} = \sup_{x \neq 0, x \in \mathbb{R}^d} \frac{\|Ax\|_p}{\|x\|_p}$. The $\ell_2$-norm (or spectral norm) $\|A\|_{(2)}$ corresponds to the maximum singular value of $A$. For a matrix $A \in \mathbb{R}^{n \times d}$, the $\ell_p$ norm of $A$ is $\|A\|_p = (\sum_{i=1}^n \sum_{j=1}^d |A_{ij}|^p)^{1/p}$, and the Frobenius norm of $A$ is $\|A\|_F = (\sum_{i=1}^n \sum_{j=1}^d A_{ij}^2)^{1/2}$. Let $A_{i:}$ be the $i$-th row of $A$, and let $A_{ij}$ be the entry in the $i$-th row and $j$-th column of $A$. Let $A^\top$ be the transpose matrix of $A$. The Singular Value Decomposition (SVD) of a matrix $A$ is $A = U\Sigma V^\top$, where $U \in \mathbb{R}^{n \times n}$ and $V \in \mathbb{R}^{d \times d}$ are orthogonal matrices, and $\Sigma \in \mathbb{R}^{n \times d}$ is a diagonal matrix containing the singular values $\sigma_1, \dots, \sigma_r$, where $r \leq \min\{n, d\}$. For a vector $x \in \mathbb{R}^n$ and a weight vector $w \in \mathbb{R}_{\geq 0}^n$, the weighted $\ell_p$-norm is $\|x\|_{w,p} = (\sum_{i=1}^n w_i|x_i|^p)^{1/p}$, and the weighted $\ell_\infty$ norm is $\|x\|_{w,\infty} = \max_{i \in [n]} |x_i|$. An $\epsilon$-net for a set $K$ in a metric space $(X, d)$ is a subset $T \subseteq K$ such that for every point $x \in K$, there exists $y \in T$ with $d(x, y) \leq \epsilon$. Given a parameter $\lambda > 0$, we define the statistical dimension of a matrix $A$ as $sd_\lambda(A) = \sum_{i=1}^r \frac{1}{1+\lambda/\sigma_i^2}$, where $r$ denotes the rank of $A$. For any vector $x \in \mathbb{R}^d$, let $\text{supp}(x) = \{i \in [d] \mid x_i \neq 0\}$ denote its support, and write $|\text{supp}(x)|$ for the number of nonzero coordinates.

$\ell_2$ **Leverage Scores.** The $\ell_2$-leverage score of the $i$-th row of a matrix $A$ is

$$\tau_{i,2}(A) = \sup_{x \in \mathbb{R}^d} \frac{|A_{i:}^\top x|^2}{\|Ax\|_2^2}.$$

Alternatively, the leverage scores can be expressed as $\tau_{i,2}(A) = \|e_i^\top U\|_2^2$, where $U \in \mathbb{R}^{n \times d}$ is an orthonormal basis for the column space of $A$ (Cohen et al., 2015). Therefore, the sum of the $\ell_2$ leverage scores satisfies $\sum_{i=1}^d \tau_{i,2}(A) = \text{rank}(A)$ (which equals $d$ when $A$ has full column rank).

## 3 SENSITIVITY SAMPLING FOR LASSO REGRESSION

In this section, we propose a sensitivity sampling algorithm for LASSO regression, called LASSO-Sens. The main goal is to derive a better upper bound on the coreset size using empirical process methods applied to the LASSO objective. The primary technical challenge lies in handling the interaction between the residual loss and the $\ell_1$ penalty, as standard empirical process techniques typically rely on analyzing the ratio between them, which is difficult to handle and may lead to better coreset size bounds. To address this issue, we provide a localization method for coresets within the empirical process framework, which decouples the problem into two components over localized regions. This allows us to analyze the empirical process in localized spaces involving only a single term. Over these localized sets we develop a weighted Gaussian empirical-process framework and derive upper bounds on the Gaussian diameter, covering numbers, and metric entropy. These ingredients yield a coreset of size $\tilde{O}(\frac{d}{\epsilon^2} \cdot (\log^3 d \min\{1, \frac{\log d}{\lambda^2}\} + \log(1/\delta)))$, which nearly matches the lower bound in the regime $\lambda = O(1/\sqrt{d})$. The detailed algorithm for constructing the LASSO coreset is given in Algorithm 1.

In sensitivity sampling, the sensitivity score of the $i$-th row for the LASSO objective is defined as

$$\varrho_i = \sup_{x \in \mathbb{R}^d} \frac{\|(Ax - b)_i\|_2^2 + \lambda \frac{\|x\|_1}{n}}{\|Ax - b\|_2^2 + \lambda \|x\|_1}, \tag{1}$$

where $\lambda > 0$. The definition of $\varrho_i$ captures the worst-case contribution to the LASSO objective, with the regularization term $\lambda \|x\|_1$ ensuring that each row contributes equally to sampling. Bounding the score $\varrho_i$ by the $\ell_2$ leverage score $\tau_i$ with an additive $1/n$ in this paper is straightforward; see Section A.1 for the formal details.

The LASSO-Sens algorithm mainly consists of a sampling procedure for coreset construction. We initialize an empty set of indices $Q$ and a zero vector $w$. Then, we calculate the sampling probability $p_i = \min\{1, \alpha(\varrho_i + \frac{1}{n})\}$, where $\alpha$ represents the over-sampling parameter. Next, the algorithm randomly selects a row index $i \in [n]$ with probability $p_i$, assigns the weight of the $i$-th row to $1/\sqrt{mp_i}$, and updates the set of indices to $Q = Q \cup \{i\}$. By repeating this sampling process $m$ times, Algorithm 1 returns the final set of row indices $Q$ and the corresponding weight vector $w$.

Before providing the theoretical guarantees for the coreset, we first present an equivalent transformation of the LASSO objective and its sensitivity scores. Let $A' = [A \ -b] \in \mathbb{R}^{n \times (d+1)}$ be the matrix obtained by concatenating $A$ and $b$, and let $x' = [x \ 1]$ be the vector obtained by concatenating $x$ with 1. Using $A'$ and $x'$, the original objective function $\min_x \|Ax - b\|_2^2 + \lambda \|x\|_1$ is rewritten as

$$\min_{x' \in \mathbb{R}^{d+1}, x'_{d+1} = 1} \|A'x'\|_2^2 + \lambda \|x'\|_1.$$

Thus, we reformulate the sensitivity score $\varrho_i$ as

$$\varrho_i = \sup_{x' \in \mathbb{R}^{d+1}, x'_{d+1} = 1} \frac{|(A'x')_i|^2 + \frac{\lambda}{n} \|x'\|_1}{\|A'x'\|_2^2 + \lambda \|x'\|_1} > 0.$$

### 3.1 SAMPLING ERROR ANALYSIS

In this subsection, we develop a localized empirical process framework to analyze the sampling error introduced by sensitivity sampling in the LASSO objective. To achieve this, we decompose

---

**Algorithm 1** LASSO-Sens

---

**Input:** a matrix $A \in \mathbb{R}^{n \times d}$, a vector $b \in \mathbb{R}^n$, a regularized parameter $\lambda$, the over-sampling parameter $\alpha$, the coreset size $m$, and a set of approximate sensitivity scores $\{\varrho_i\}_{i=1}^n$
**Output:** a set of indices $Q$, and a weight vector $w \in \mathbb{R}_{\geq 0}^n$
  1: Initialize an empty set $Q$, and let $w$ be an $n$-dimensional zero vector.
  2: **for** $i \leftarrow 1, 2, \ldots, n$ **do**
  3:     Compute the sampling probability for the $i$-th row $p_i = \min\{1, \alpha(\varrho_i + \frac{1}{n})\}$.
  4: **end for**
  5: **for** $t \leftarrow 1, 2, \ldots, m$ **do**
  6:     Sample a row index $i \in [n]$ with probability $p_i$, and set the weight $w_i = 1/\sqrt{mp_i}$.
  7:     $Q \leftarrow Q \cup \{i\}$.
  8: **end for**
  9: **return** $Q$ and $w$.

---

the function space into a residual component and a penalty component, and localize our study to their intersection. This separation enables us to independently bound the Gaussian complexity and metric entropy of each component using a combination of weighted chaining techniques. By constructing multi-scale $\epsilon$-nets and applying concentration inequalities for Gaussian processes, we establish an upper bound on the coreset size that controls the sampling error.

We now analyze the sampling error introduced by sensitivity sampling. Let $\{p_i\}_{i=1}^n$ denote the sampling probabilities associated with each row of the augmented matrix $A'$. Let $S \in \{0,1\}^{m \times n}$ be a sampling matrix with exactly one nonzero entry in each row, where $m$ represents the number of rows sampled from $A'$. Let $w$ be the weight vector corresponding to the selected rows. Let $\mathcal{T} = \{x \mid x \in \mathbb{R}^{d+1}, x_{d+1} \neq 0\}$, and let $\Omega = \{x \mid x \in \mathcal{T}, \|A'x\|_2^2 + \lambda\|x\|_1 = 1\}$ be the unit ball of the LASSO objective. Then, we define the sampling error $\mathcal{E}$ over the domain $\Omega$ as

$$\mathcal{E} = \sup_{x' \in \Omega} \left| \|SA'x'\|_{w,2}^2 + \lambda\|x'\|_1 - (\|A'x'\|_2^2 + \lambda\|x'\|_1) \right|$$
$$= \sup_{x' \in \Omega} \left| \|SA'x'\|_{w,2}^2 - \|A'x'\|_2^2 \right|.$$

Our goal is to bound $\mathcal{E}$ by $\epsilon$, leading to the inequality

$$\|SA'x'\|_{w,2}^2 + \lambda\|x'\|_1 \leq (1 \pm \epsilon)\left(\|A'x'\|_2^2 + \lambda\|x'\|_1\right)$$

for every $x' \in \Omega$. To bound $\mathcal{E}$ using the chaining method (Cohen & Peng, 2015; Koltchinskii, 2001; Hu et al., 2022), we analyze the moments of $\mathcal{E}$ with the symmetrization technique, which allows us to construct a Gaussian reduction as follows. (A detailed proof of Lemma 1 is given in Appendix Lemma 11.)

**Lemma 1.** *Let $A' \in \mathbb{R}^{n \times (d+1)}$, let $S$ be a random sampling matrix, and let $Q$ denote the set of sampled rows from $A'$. For $\lambda > 0$ and an integer $l \geq 2$, the following inequality holds*

$$\mathbb{E}_S |\mathcal{E}|^l \leq (2\pi)^{l/2} \mathbb{E}_S \mathbb{E}_{g \sim \mathcal{N}(0, I_n)} \sup_{x \in \Omega} \left| \sum_{i \in Q} g_i w_i |(A_{i:} x)|^2 \right|^l,$$

*where $g \sim \mathcal{N}(0, I_n)$ represents a Gaussian vector with independent entries.*

We bound the sampling error $\mathcal{E}$ by analyzing the associated Gaussian process, as described in Lemma 1. To handle higher-order moments of $\mathcal{E}$, we apply a moment bound from Woodruff & Yasuda (2023), which uses Dudley's tail inequality for Gaussian processes. Consequently, we obtain the following inequality for the sampling error.

$$\mathbb{E}_S[|\mathcal{E}|^l] \leq (C\mathcal{M}_\mathcal{E})^l (\mathcal{M}_\mathcal{E}/\mathcal{D}) + O(\sqrt{l}\mathcal{D})^l, \tag{2}$$

where $C$ is an absolute constant, $\mathcal{M}_\mathcal{E}$ denotes the metric entropy of the Gaussian process, and $\mathcal{D}$ is the Gaussian diameter. (The detailed definitions of $\mathcal{M}_\mathcal{E}$ and $\mathcal{D}$ are provided in the following.)

By appropriately choosing the parameter $l$ and bounding both the metric entropy $\mathcal{M}_\mathcal{E}$ and the Gaussian diameter $\mathcal{D}$ of the Gaussian process, we can ensure that $\mathbb{E}_S[|\mathcal{E}|^l] \leq \epsilon^l$, which leads to a sufficiently small coreset size $m$. We now decompose the unit ball $\mathcal{L} = \{x \mid x \in \mathcal{T}, \|A'x\|_2^2 + \lambda\|x\|_1 \leq$

1}, which arises from the residual term $\|A'x\|_2$ and the $\ell_1$ penalty term. (The proof of Lemma 2 is provided in Appendix Lemma 12.)

**Lemma 2.** *Let $A' \in \mathbb{R}^{n \times (d+1)}$ be a matrix and $\lambda > 0$. Define the sets $\Omega = \{x \mid x \in \mathcal{T}, \|A'x\|_2^2 + \lambda\|x\|_1 \le 1\}$ and $\mathcal{L} = \{x \mid x \in \mathcal{T}, \|A'x\|_2^2 \le 1 \text{ and } \|x\|_1 \le \frac{1}{\lambda}\}$. Then, it holds that $\Omega \subseteq \mathcal{L}$.*

Define $B_2(A') = \{x \mid x \in \mathcal{T}, \|A'x\|_2^2 \le 1\}$ as the unit ball in the residual space, and $B_1(\frac{1}{\lambda}) = \{x \mid x \in \mathcal{T}, \|x\|_1 \le \frac{1}{\lambda}\}$ as the unit ball in the $\ell_1$-penalty space. By Lemma 2, we have

$$\mathcal{L} \subseteq \mathcal{L}_{A'} = B_2(A') \cap B_1(\frac{1}{\lambda}).$$

This allows us to proceed with bounding both the Gaussian diameter $\mathcal{D}$ and the metric entropy $\mathcal{M}_{\mathcal{E}}$ for the convex sets $B_2(A')$ and $B_1\left(\frac{1}{\lambda}\right)$, respectively. Let $M = SA'$, where $S \in \{0,1\}^{m \times n}$ is a sampling matrix.

We start by bounding the Gaussian diameter $\mathcal{D}$ by relaxing the pseudo-metric $d_X$ using an upper bound involving the maximum $\ell_2$ leverage score and $\lambda$. Define the convex set $\mathcal{L}_M = \{y = Mx \mid x \in \mathcal{L}_{A'}\}$. Let $\tau = \sup_{x' \in \Omega, i \in [n]} |A'_{i:}x'|^2$ be the maximum row contribution (under the normalization $x' \in \Omega$). Next, we prove that the diameter $\mathcal{D}(\mathcal{L}_M)$ with respect to $d_X$ is bounded by the following inequality. (Detailed proof of Lemma 3 is given in Appendix Lemma 13.)

**Lemma 3.** *Let $M \in \mathbb{R}^{m \times (d+1)}$, and let $w$ be the weight vector. Define the pseudo-metric*

$$d_X(y, y') = \left(\mathbb{E}_{g \sim \mathcal{N}(0, I_n)} \left| \sum_{i=1}^{m} g_i w_i |y_i|^2 - \sum_{i=1}^{m} g_i w_i |y'_i|^2 \right|^2 \right)^{1/2}$$

*for any $y, y' \in \mathcal{L}_M$. Then, the diameter $\mathcal{D}(\mathcal{L}_M)$ with respect to $d_X$ is bounded by*

$$\mathcal{D}(\mathcal{L}_M) \le O(\tau\sqrt{\log d}/\lambda).$$

To obtain a precise bound for the Gaussian process over $\mathcal{L}_M$, we apply the chaining method to construct a sequence of $t$-nets at varying scales $t > 0$, which capture the convex structure of the function class associated with $\mathcal{E}$ on $\mathcal{L}_M$. Utilizing this chaining method, we can derive a bound on $\mathbb{E}_S|\mathcal{E}|^l$ via the covering numbers of the sequence of $t$-nets. Thus, we aim to bound the minimal number of weighted unit $\ell_p$ (or $\ell_\infty$) balls required to cover the convex set $\mathcal{L}_M$ for $p \in [1, \infty)$. We define the weighted unit ball of the residual space $B_{w,2}(M)$ as $B_{w,2}(M) = \{x \mid x \in \mathcal{T}, \|Mx\|_{w,2} \le 1\}$, and define $\mathcal{L}_{w,M} = \{y = Mx \mid x \in B_{w,2}(M) \cap B_1(1/\lambda)\}$. Let $G = 1 + \mathcal{E} = 1 + \sup_{x' \in \mathcal{L}_M} \left| \|SA'x'\|_{w,2}^2 - \|A'x'\|_2^2 \right|$.

To bound the metric entropy of the convex set $B_{w,2}(M)$, we first define the weighted $\ell_p$-unit ball as $B_{w,p}(M) = \{x \mid x \in \mathcal{T}, \|Mx\|_{w,p} \le 1\}$. Let $\mathcal{T}_p$ denote a $t$-net of $B_{w,2}(M)$ with respect to the weighted $\ell_p$-norm, i.e., a finite subset of $B_{w,2}(M)$ such that every point in $B_{w,2}(M)$ is within distance $t$ (measured in $\|\cdot\|_{w,p}$) from some point in $\mathcal{T}_p$. We define $N(B_{w,2}(M), \|\cdot\|_{w,p}, t)$ as the minimal cardinality of such a set $\mathcal{T}_p$, and the metric entropy of $B_{w,2}(M)$ with respect to the weighted $\ell_p$-norm is then defined as $\log N(B_{w,2}(M), \|\cdot\|_{w,p}, t)$. (Detailed definitions are provided in Appendix Definition 14 and Definition 15.)

**Lemma 4** ((Munteanu & Omlor, 2024), slightly modified). *Let $2 \le p < \infty$, and let $M \in \mathbb{R}^{m \times (d+1)}$ be an orthonormal matrix with a weight vector $w \in \mathbb{R}_{\ge 0}^m$. Then, the following inequalities hold*

$$\log N(B_{w,2}(M), \|\cdot\|_{w,p}, t) \le O(1)\frac{m^{2/p} p \cdot \tau}{t^2} \text{ and } \log N(B_{w,2}(M), \|\cdot\|_{w,\infty}, t) \le O(1)\frac{\log m \cdot \tau}{t^2}.$$

For bounding the metric entropy of the convex set $B_1\left(\frac{1}{\lambda}\right)$, we aim to bound the number of $B_\infty$-balls of radius $t$ needed to cover the $B_1$-ball. Specifically, the covering process can be decomposed into two steps: first, cover the $B_2$-ball using $B_\infty$-balls, and second, cover the $B_1$-ball using $B_2$-balls. The $B_1$ ball has a unique geometric structure, with a large portion of its volume concentrated near its center, as pointed out in (Vershynin, 2018). This concentration implies that fewer small-radius balls are required to cover $B_1$, compared to naive volume-based estimates. While a straightforward volumetric argument yields a worst-case covering number of $O((1 + \frac{1}{\epsilon})^d)$, this bound can be quite loose. To obtain a tighter estimate, we leverage the Sudakov minoration inequality (Vershynin, 2018), which provides a way to control the covering number $N(B_1, B_\infty, t)$. (A Detailed proof of Lemma 5 is given in Appendix Lemma 21.)

**Lemma 5.** *Let $p \geq 1$ be a parameter, and let $B_p = \{x \in \mathbb{R}^d \mid \|x\|_p \leq 1\}$ be the unit ball for the $\ell_p$ norm. Then, $\log N(B_1, B_\infty, t) \leq O(\frac{\log d}{t})$.*

To bound the metric entropy of these $t$-nets, we need to calculate the following integral

$$\mathcal{M}_\mathcal{E} \leq \int_0^\infty \sqrt{\log N(\mathcal{L}_{w,M}, d_X, t)} \, dt.$$

For radii $t > \mathcal{D}(\mathcal{L}_{w,M})$, the covering number satisfies $\log N(\mathcal{L}_{w,M}, d_X, t) = 0$, which implies that any single vector $y \in \mathcal{L}_{w,M}$ serves as a $t$-net. Therefore, we only need to focus on the case where the radius $t$ lies within the interval $[0, \mathcal{D}(\mathcal{L}_{w,M})]$. We derive the following inequality, whose proof provided in Appendix Lemma 27.

**Lemma 6.** *Let $M \in \mathbb{R}^{m \times (d+1)}$ be a matrix and let $\lambda$ be a positive parameter. Then, the metric entropy $\mathcal{M}_\mathcal{E}$ of $\mathcal{L}_{w,M}$ satisfies*

$$\int_0^\infty \sqrt{\log N(\mathcal{L}_{w,M}, d_X, t)} \, dt \leq O\left(G\tau\sqrt{\log d}\right)\left(1 + \log m \cdot \min\left\{1, \frac{\sqrt{\log d}}{\lambda}\right\}\right),$$

*where $\tau$ is the maximum weighted $\ell_2$-leverage score of $M$.*

We now present the main result, which provides a bound on the coreset size required to ensure that $\mathbb{E}|\mathcal{E}|^l \leq \epsilon$. (The proof of Theorem 7 is provided in Appendix Theorem 30.)

Table 1: Comparison results of loss, runtime, and sparsity on CTs dataset ($n = 53,500$, $d = 386$) for varying coreset sizes at $\lambda = 0.5$.

| Metrics | Algorithms | Coreset Sizes | | | | | |
|---|---|---|---|---|---|---|---|
| | | $1d$ | $2d$ | $5d$ | $10d$ | $15d$ | $20d$ |
| Loss | LASSO | | | **25.28±0.13** | | | |
| | **LASSO-Sens** | 29.27±7.05 | 25.43±0.17 | 25.34±0.03 | 25.32±0.02 | 25.32±0.01 | 25.32±0.02 |
| | LASSO-Uniform | 61.58±22.86 | 30.39±6.19 | 25.94±0.42 | 25.58±0.17 | 25.55±0.13 | 25.46±0.13 |
| Time (s) | LASSO | | | 505.65 | | | |
| | **LASSO-Sens** | 9.07 | 13.75 | 20.84 | 34.58 | 66.40 | 113.34 |
| | LASSO-Uniform | 8.70 | 13.41 | 21.12 | 36.96 | 72.74 | 122.72 |
| Sparsity | LASSO | | | 325 | | | |
| | **LASSO-Sens** | 381 | 360 | 336 | 329 | **322** | **318** |
| | LASSO-Uniform | 379 | 360 | 343 | 337 | 329 | 325 |

**Theorem 7.** *Let $A' \in \mathbb{R}^{n \times (d+1)}$ be an input matrix, let $S$ be a random sampling matrix, and let $\varepsilon, \delta \in (0, 1)$ and $\lambda > 0$ be a parameters. If $\alpha = \tilde{O}\left(\frac{1}{\epsilon^2} \cdot \left(\log\left(d\log(1/\delta)\right)(\ln d)^2 \cdot \min\left\{1, \frac{\log d}{\lambda^2}\right\} + \log\frac{1}{\delta}\right)\right)$ and for all $i \in [n]$ it holds that*

$$p_i \geq \min\{1, \alpha(\tau_{i,2}(A') + \frac{1}{n})\},$$

*where $\tau_{i,2}(A')$ denotes the $\ell_2$ leverage score of the $i$-th row of $A'$. Then, with failure probability at most $\delta$, it holds that for all $x \in \mathbb{R}^{d+1}$ with $x_{d+1} = 1$,*

$$\|SA'x\|_{w,2}^2 + \lambda\|x\|_1 \leq (1 \pm \epsilon)(\|A'x\|_2^2 + \lambda\|x\|_1),$$

*and the coreset size is at most $m = \tilde{O}\left(\frac{d(\log d)^3}{\epsilon^2} \cdot \min\{1, \frac{\log d}{\lambda^2}\} + \frac{d}{\epsilon^2}\log\frac{1}{\delta}\right)$.*

To establish a lower bound on the coreset size $m$, we utilize a reduction from the support recovery problem in sparse recovery. We consider the task of recovering the support of a sparse vector $x^*$, and apply information-theoretic techniques to LASSO regression. Our analysis shows that, under certain conditions, any algorithm achieving a $(1 + \epsilon)$-approximation from the coreset requires at least $\Omega(\frac{d}{\epsilon^2}\log d)$ rows. Since Mai et al. (2023) pointed out the lack of scale-invariance in LASSO objective, we normalize the inputs by assuming $\|A\|_2 \leq 1$ and $\|b\|_2 \leq 1$. Detailed proofs are provided in Section B.4.

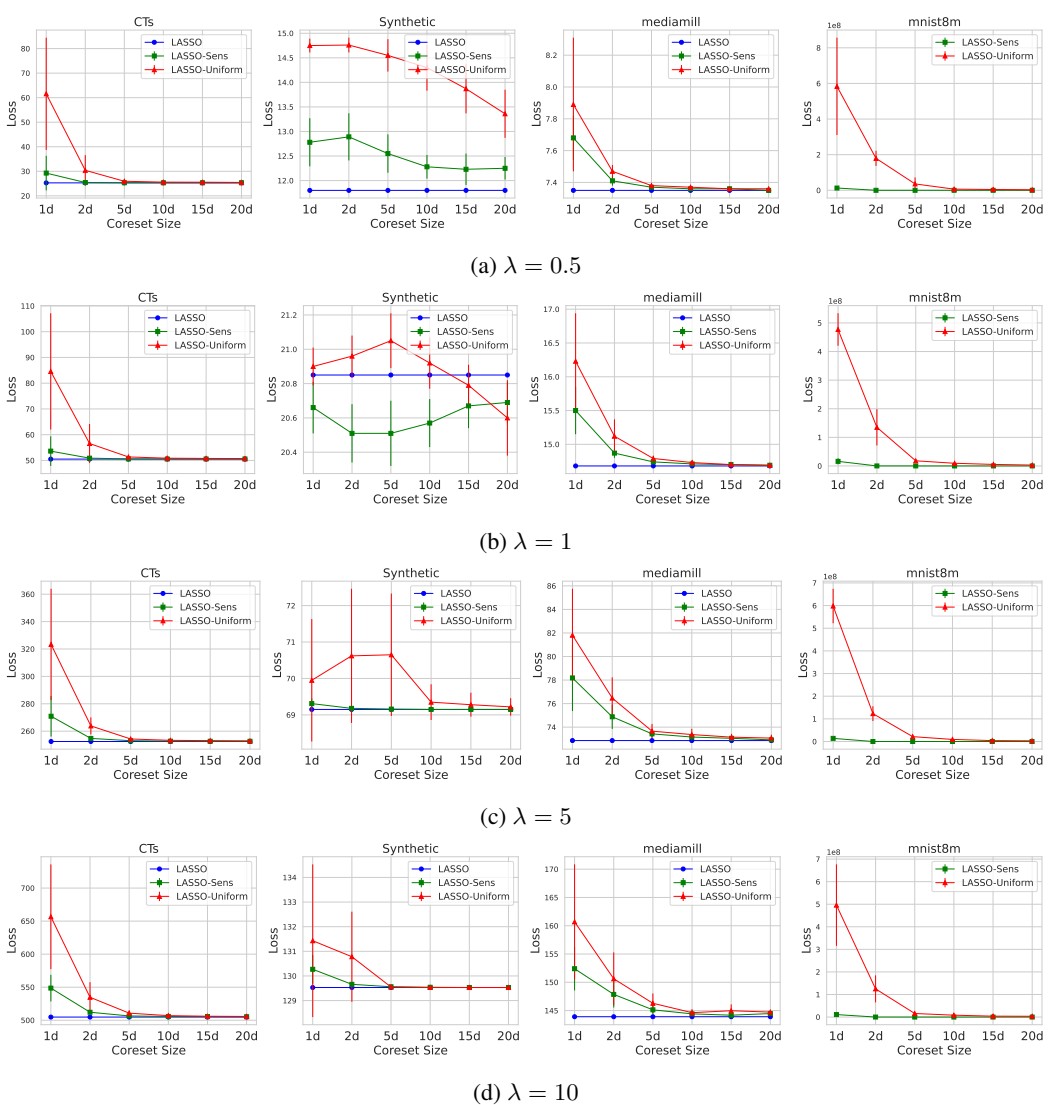

(a) $\lambda = 0.5$

(b) $\lambda = 1$

(c) $\lambda = 5$

(d) $\lambda = 10$

Figure 1: LASSO regression loss comparison across varying coreset sizes for $\lambda = \{0.5, 1, 5, 10\}$.

**Lemma 8.** *Let* $A \in \mathbb{R}^{n \times d}$, $b \in \mathbb{R}^n$, *and* $\lambda \in (0, 1)$. *Assume that* $\|A\|_2 \le 1$ *and* $\|b\|_2 \le 1$. *Let* $S$ *be a diagonal sampling matrix with* $m$ *nonzero entries. Suppose there exists an estimator whose output* $\tilde{x} = \arg\min_{x \in \mathbb{R}^d} \|SAx - Sb\|_2^2 + \lambda \|x\|_1$ *satisfies*

$$\|A\tilde{x} - b\|_2^2 + \lambda \|\tilde{x}\|_1 \le (1 + \epsilon) \cdot \min_{x \in \mathbb{R}^d} (\|Ax - b\|_2^2 + \lambda \|x\|_1).$$

*Then, the coreset size* $m$ *must satisfy*

$$m = \begin{cases} \Omega(\frac{\log d}{\lambda^2 \cdot \epsilon^2}), & \text{if } \lambda = \Omega(\frac{1}{\sqrt{d}}) \\ \Omega(\frac{d}{\epsilon^2} \log d), & \text{if } \lambda = O(\frac{1}{\sqrt{d}}) \end{cases}.$$

## 4 EXPERIMENTS

In this section, we compare three algorithms for solving the LASSO regression problem: direct optimization using the full dataset, and solving LASSO on subsamples selected via sensitivity sampling and uniform sampling, respectively. All experiments are conducted on a machine with 72 Intel Xeon Gold 6230 CPUs and 340 GB of memory, and all implementations are executed in MATLAB 2017a.

**Datasets.** We evaluate the three algorithms on 4 datasets: Synthetic ($n = 10,000, d = 200$), mediamill ($n = 30,993, d = 120$), CTs ($n = 53,500, d = 386$), and mnist8m ($n = 8,100,000, d = 784$). The synthetic dataset is generated by constructing a matrix $A \in \mathbb{R}^{10000 \times 200}$, where a small number of rows have high leverage scores. This construction follows the method described in (Chhaya et al., 2020). The resulting matrix is designed to exhibit a non-uniform leverage score distribution while maintaining a well-conditioned structure. For all datasets, the response vector is defined as $b = Ax + 10^{-5} \cdot \frac{\|b\|_2}{\|e\|_2} \cdot e$, where $x \in \{0,1\}^d$ is a randomly generated sparse vector, and $e$ is a noise vector. All datasets used in our experiments are publicly available at the following URLs: https://archive.ics.uci.edu/datasets and https://www.csie.ntu.edu.tw/~cjlin/libsvmtools/datasets/.

**Algorithms.** In our experimental evaluation, we compare the following three algorithms:

- LASSO. The standard LASSO regression is solved using the FISTA method as described in (Beck & Teboulle, 2009).

- LASSO-Sens. Our proposed approach (see Algorithm 1), which first constructs a coreset via sensitivity-based sampling and then solves the LASSO problem on the coreset using FISTA.

- LASSO-Uniform. A baseline that first uniformly samples rows from the input data and then applies FISTA to solve the LASSO problem on the sampled data.

**Methodology.** We evaluate algorithm performance using the loss function $f(x) = \|Ax - b\|_2^2 + \lambda\|x\|_1$, where a lower value of the loss indicates a better solution. To evaluate the sparsity of the solution, we follow the method introduced in (Chhaya et al., 2020) by setting any entry of $x$ with an absolute value less than $10^{-6}$ to 0, and we count the remaining nonzero entries. Our experiments test three methods: LASSO, LASSO-Sens, and LASSO-Uniform on four datasets. To ensure a fair comparison, we test each algorithm 10 times and report the average loss, runtime, and sparsity. To compare the performance of different sampling strategies, we run LASSO-Sens and LASSO-Uniform across a range of coreset sizes and regularization parameters, with the values $\lambda \in \{0.5, 1, 5, 10\}$. Specifically, the coreset size is selected from $\{1, 2, 5, 10, 15, 20\} \times d$ for each dataset.

**Results for the LASSO Regression.** As shown in Figure 1, the LASSO-Sens algorithm achieves loss values that closely match those of the exact LASSO solver as the coreset size increases, particularly for $\lambda = 0.5$ and $\lambda = 10$. The comparison of performance metrics across four datasets under varying values of $\lambda$ and coreset sizes is reported in Table 1 and Appendix Tables 2-5, including average loss, standard deviation, runtime, and solution sparsity. On the Synthetic, Mediamill, and CTs datasets, LASSO-Sens is at least 4 times faster than LASSO, and up to 18 times faster on CTs. On the mnist8m dataset, LASSO-Sens obtains a feasible solution within 15 minutes, whereas the standard LASSO solver fails to return a solution even after 48 hours. Furthermore, the LASSO-Sens algorithm generally outperforms LASSO-Uniform in terms of both accuracy and sparsity on the mnist8m dataset. At a coreset size of $10d$, the sparsity of the solutions produced by LASSO-Sens closely matches that of the exact LASSO solver across all datasets. These experimental results show the sensitivity sampling in accelerating the LASSO regression process while preserving high-quality solutions and solution sparsity. All of these findings, together with our Theorem 7, confirm the effectiveness of sensitivity sampling for LASSO.

## 5 Conclusion

In this paper, we propose the first coreset construction method for LASSO regression via a sensitivity sampling algorithm. Directly applying existing coreset techniques for regularized regression to LASSO yields a coreset size bound of $\tilde{O}(\mathcal{G}d/\epsilon^2)$. To achieve a smaller coreset, we propose an empirical process analysis that addresses the complex function-space arising from the interaction between the residual error and the $\ell_1$ penalty in LASSO objective, thereby achieving a coreset of size $\tilde{O}(\epsilon^{-2}d \cdot (\log^3 d \cdot \min\{1, \log d/\lambda^2\} + \log(1/\delta)))$. An interesting future direction is to study how our method can be extended to the elastic net and other regression problems involving more complex regularization.

## ACKNOWLEDGEMENTS

This work was supported by National Natural Science Foundation of China (62432016, 62502545), and Central South University Research Programme of Advanced Interdisciplinary Studies (2023QYJC023). This work was also carried out in part using computing resources at the High Performance Computing Center of Central South University.

## ETHICS STATEMENT

This work does not involve any human subjects, sensitive data, or other ethical concerns as outlined in the ICLR Code of Ethics.

## REPRODUCIBILITY STATEMENT

This paper is committed to ensuring the reproducibility of our work. To ensure the completeness of the comparison, we have provided detailed descriptions of our proposed method and its components in Section 4 of the main paper. To enable accurate replication, we clearly specify all hyperparameters, training procedures, and evaluation protocols in the same section. Additional implementation results, including figures and tables, are provided in the Appendix.

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

## A  APPENDIX

### A.1  MISSING PROOF OF SENSITIVITY SCORES

In this subsection, we provide an upper bound on the sensitivity score $\rho_i$ using the $\ell_2$ leverage score and a fixed term $1/n$. While this result is not fundamentally new (see, e.g., (Mahoney et al., 2011; Woodruff, 2014; Chhaya et al., 2020)), we slightly extend the well-conditioned basis method to the LASSO objective.

**Definition 9.** ($\ell_2$ *Well-Conditioned Basis.*) *Given a matrix* $A \in \mathbb{R}^{n \times d}$, *we define a* $(\sqrt{d}, 1, 2)$ *well-conditioned basis for* $A$ *such that* $\|U\|_2 \leq \sqrt{d}$, *and for all* $x \in \mathbb{R}^d$, $\|x\|_2 \leq \|Ux\|_2$, *where* $U \in \mathbb{R}^{n \times d}$ *is the orthogonal matrix obtained from the SVD of* $A$.

**Lemma 10.** *Let* $A' \in \mathbb{R}^{n \times (d+1)}$, *and let* $\lambda > 0$ *be a regularization parameter. Then, the estimated sensitivity score* $\hat{\varrho}_i$ *satisfies* $\hat{\varrho}_i = 2\tau_{i,2}(A') + \frac{1}{n} \geq \varrho_i$, *where* $\tau_{i,2}(A')$ *denotes the* $\ell_2$ *leverage score of the* $i$-*th row of* $A'$. *All sensitivity scores* $\hat{\varrho}_i$ *can be computed in time* $O(nnz(A') \log n + d^3 \log(n/d) \log d)$, *where* $nnz(A')$ *denotes the number of nonzero entries in* $A'$. *Moreover, the total sensitivity is bounded as* $\mathcal{G} \leq 2d + 3$.

*Proof.* Let $A' = UV$, where $U \in \mathbb{R}^{n \times (d+1)}$ is a $(\sqrt{d+1}, 1, 2)$-well-conditioned basis for $A'$. Denote the $i$-th row of $A'$ as $A'_i = u_i^\top V$, where $u_i^\top$ is the $i$-th row of $U$. For any $x' \in \mathbb{R}^{d+1}$, define $z = Vx'$, so that $A'x' = Uz$. Let $T = \{x' \in \mathbb{R}^{d+1} : x'_{d+1} = 1\}$. Then, we obtain

$$\varrho_i = \sup_{x' \in T} \frac{|A'_i x'|^2 + \frac{\lambda}{n}\|x'\|_1}{\|A'x'\|_2^2 + \lambda\|x'\|_1} = \sup_z \frac{|u_i^T z|^2 + \frac{\lambda}{n}\|V^{-1}z\|_1}{\|Uz\|_2^2 + \lambda\|V^{-1}z\|_1} \leq \sup_z \frac{|u_i^T z|^2}{\|Uz\|_2^2} + \frac{1}{n} \leq \tau_{i,2}(A') + \frac{1}{n}.$$

Thus, the total sensitivity satisfies $\mathcal{G} = \sum_{i=1}^n \varrho_i \leq \sum_{i=1}^n \left(\tau_{i,2}(A') + \frac{1}{n}\right) \leq d + 2$, where $\tau_{i,2}(A') = \|u_i\|_2^2$ denotes the $\ell_2$ leverage score of the $i$-th row. Furthermore, by extending Lemma 8 of (Cohen et al., 2015), the approximate leverage score $\hat{\tau}_{i,2}(A') \leq 2\tau_{i,2}(A')$ can be computed in time $O(nnz(A') \log n + d^3 \log d \log(n/d))$. Substituting this into the bound yields $\varrho_i \leq 2\tau_{i,2}(A') + \frac{1}{n}$ and $\mathcal{G} \leq 2d + 3$. $\qquad\square$

## B  OMITTED PROOFS OF SAMPLING ERROR ANALYSIS

In this section, we reduce the empirical process associated with the sampling error $\mathcal{E}$ to a Gaussian process using the symmetrization technique. The sampling error $\mathcal{E}$ is defined on the set $\Omega = \{x \in \mathcal{T} \mid \|A'x\|_2^2 + \lambda\|x\|_1 = 1\}$, where $\mathcal{T} = \{x \in \mathbb{R}^{d+1} \mid x_{d+1} = 1\}$. To analyze the functional complexity, we consider the larger set $\mathcal{T}' = \{x \in \mathbb{R}^{d+1} \mid x_{d+1} \neq 0\}$, since any $x \in \mathcal{T}$ can be obtained by scaling an element of $\mathcal{T}'$. Specifically, for each $x \in \mathcal{T}$, there exists a scalar $c$ and a vector $x' \in \mathcal{T}'$ such that $x = c \cdot x'$, which implies $\mathcal{T} \subseteq \mathcal{T}'$. Consequently, we define the extended domain $\Omega' = \{x \in \mathcal{T}' \mid \|A'x\|_2^2 + \lambda\|x\|_1 = 1\}$ and focus our subsequent analysis on this set using tools from Gaussian process theory, particularly those developed for unregularized regression in Woodruff & Yasuda (2023).

**Lemma 11.** *Let* $A' \in \mathbb{R}^{n \times (d+1)}$, *let* $S$ *be a random sampling matrix, and let* $Q$ *denote the set of sampled rows from* $A'$. *For* $\lambda > 0$ *and an integer* $l \geq 2$, *the following inequality holds*

$$\mathbb{E}_S |\mathcal{E}|^l \leq (2\pi)^{l/2} \mathbb{E}_S \mathbb{E}_{g \sim \mathcal{N}(0, I_n)} \sup_{x \in \Omega} \left| \sum_{i \in Q} g_i w_i |(A_{i:}x)|^2 \right|^l,$$

*where* $g \sim \mathcal{N}(0, I_n)$ *represents a Gaussian vector with independent entries.*

*Proof.* We consider the simple convex function $|a + b|^l$ for $a, b \in \mathbb{R}_+$, where $l > 1$ is a positive number. Given a random sampling matrix $S$, the linearity of expectation implies

$$\mathbb{E}\left[\|SA'x\|_{w,2}^2 + \lambda\|x\|_1\right] = \|A'x\|_2^2 + \lambda\|x\|_1$$

for any vector $x \in \mathbb{R}^{d+1}$. Next, without loss of generality, we assume that $\|A'x\|_2^2 + \lambda\|x\|_1 = 1$; otherwise, we can rescale $x$ by a constant to satisfy this condition.

We now analyze the following quantity

$$\mathcal{E} = \mathbb{E}_S \sup_{\|A'x\|_2^2 + \lambda\|x\|_1 = 1, x \in \mathcal{T}} \left| \|SA'x\|_{w,2}^2 + \lambda\|x\|_1 - 1 \right|$$

Let $S'$ be an independently copy of $S$. Applying Jensen inequality, we have

$$\mathbb{E}_S \sup_{\|A'x\|_2^2 + \lambda\|x\|_1 = 1, x \in \mathcal{T}} |\|SA'x\|_{w,2}^2 + \lambda\|x\|_1 - (\|A'x\|_2^2 + \lambda\|x\|_1)|^l$$

$$= \mathbb{E}_S \sup_{\|A'x\|_2^2 + \lambda\|x\|_1 = 1, x \in \mathcal{T}} \left| \|SA'x\|_{w,2}^2 - \|A'x\|_2^2 + 0 \right|^l$$

$$= \mathbb{E}_S \sup_{\|A'x\|_2^2 + \lambda\|x\|_1 = 1, x \in \mathcal{T}} |\|SA'x\|_{w,2}^2 - \|A'x\|_2^2 + \mathbb{E}_{S'}(\|A'x\|_2^2 - \|S'A'x\|_{w,2}^2)|^l$$

$$\leq \mathbb{E}_{S,S'} \sup_{\|A'x\|_2^2 + \lambda\|x\|_1 = 1, x \in \mathcal{T}} \left| \|\|SA'x\|_{w,2}^2 - \mathbb{E}_{S'}\|S'A'x\|_{w,2}^2 \right|^l$$

$$\leq \mathbb{E}_{S,S'} \sup_{\|A'x\|_2^2 + \lambda\|x\|_1 = 1, x \in \mathcal{T}} \left| \|\|SA'x\|_{w,2}^2 - \|S'A'x\|_{w,2}^2 \right|^l.$$

Using a standard symmetrization argument (Vershynin, 2018), we obtain

$$\mathbb{E}_{S,S'} \sup_{\|A'x\|_2^2 + \lambda\|x\|_1 = 1, x \in \mathcal{T}} \left| \|SA'x\|_{w,2}^2 - \|S'A'x\|_{w,2}^2 \right|^l$$

$$\leq 2^l \mathbb{E}_{S,\epsilon} \sup_{\|A'x\|_2^2 + \lambda\|x\|_1 = 1, x \in \mathcal{T}} \left| \sum_{i \in Q} \epsilon_i w_i |(A'_{i:})x|^2 \right|^l$$

$$\leq 2^l (\pi/2)^{l/2} \mathbb{E}_{S,g} \sup_{\|A'x\|_2^2 + \lambda\|x\|_1 = 1, x \in \mathcal{T}} \left| \sum_{i \in Q} g_i w_i |(A'_{i:})x|^2 \right|^l,$$

where $\epsilon \sim \{\pm 1\}^n$ are independent Rademacher variables in the first inequality, and the second inequality follows from the Rademacher–Gaussian comparison theorem (Ledoux & Talagrand, 1991) with $g \sim \mathcal{N}(0, I_n)$ a standard Gaussian vector in $\mathbb{R}^n$. □

**Lemma 12.** *Let $A' \in \mathbb{R}^{n \times (d+1)}$ be a matrix and $\lambda > 0$. Define the sets $\Omega = \{x \mid x \in \mathcal{T}, \|A'x\|_2^2 + \lambda\|x\|_1 \leq 1\}$ and $\mathcal{L} = \{x \mid x \in \mathcal{T}, \|A'x\|_2^2 \leq 1 \text{ and } \|x\|_1 \leq \frac{1}{\lambda}\}$. Then, it holds that $\Omega \subseteq \mathcal{L}$.*

*Proof.* Take any $x \in \Omega$. By the definition of the set $\Omega$, we have

$$\|A'x\|_2^2 + \lambda\|x\|_1 \leq 1.$$

Since $\|A'x\|_2^2$ is non-negative, it follows that

$$\lambda\|x\|_1 \leq \|A'x\|_2^2 + \lambda\|x\|_1 \leq 1,$$

hence

$$\|x\|_1 \leq \frac{1}{\lambda}.$$

Next, since $\lambda\|x\|_1 \geq 0$, we have

$$\|A'x\|_2^2 \leq \|A'x\|_2^2 + \lambda\|x\|_1 \leq 1.$$

Thus, we have

$$\|A'x\|_2^2 \leq 1.$$

Since $x \in \Omega$ implies $x \in \mathcal{T}$ by definition, we conclude that $x \in \mathcal{L}$. Since $x$ was arbitrary, we have $\Omega \subseteq \mathcal{L}$. □

### B.1 BOUNDING THE GAUSSIAN DIAMETER

We start by bounding the Gaussian diameter $\mathcal{D}$ with respect to the pseudo-metric $d_X$.

**Lemma 13.** *Let $M \in \mathbb{R}^{m \times (d+1)}$, and let $w$ be a weight vector. Define the pseudo-metric*

$$d_X(y, y') = \left( \mathbb{E}_{g \sim \mathcal{N}(0, I_n)} \left| \sum_{i=1}^{m} g_i w_i |y_i|^2 - \sum_{i=1}^{m} g_i w_i |y_i'|^2 \right|^2 \right)^{1/2}$$

*for any $y, y' \in \mathcal{L}_M$. Then, the diameter $\mathcal{D}(\mathcal{L}_M)$ with respect to $d_X$ is bounded by*

$$\mathcal{D}(\mathcal{L}_M) \leq O(\tau \sqrt{\log d}/\lambda).$$

*Proof.* Let $W = \mathrm{diag}(w)$ and recall the pseudo-metric

$$d_X(y, y') := \left( \mathbb{E}_{g \sim \mathcal{N}(0, I_m)} \left| \sum_{i=1}^{m} g_i w_i (|y_i|^2 - |y_i'|^2) \right|^2 \right)^{1/2}, \text{where } y, y' \in \mathcal{L}_M.$$

Since $g \sim \mathcal{N}(0, I_m)$ and the coordinates are independent, we have

$$d_X(y, y')^2 = \sum_{i=1}^{m} w_i^2 \left( |y_i|^2 - |y_i'|^2 \right)^2, \qquad \text{hence} \qquad d_X(y, y') = \| w \odot (|y|^2 - |y'|^2) \|_2.$$

For each coordinate, using $|a^2 - b^2| = |a - b||a + b|$ and $||y_i| - |y_i'|| \leq |y_i - y_i'|$, we obtain

$$\left| |y_i|^2 - |y_i'|^2 \right| \leq (|y_i| + |y_i'|) |y_i - y_i'|.$$

Then $|y_i|, |y_i'| \leq \sqrt{\tau}$ for $y, y' \in \mathcal{L}_M$, we get

$$\left| |y_i|^2 - |y_i'|^2 \right| \leq 2\sqrt{\tau} |y_i - y_i'|.$$

Plugging this into the expression of $d_X$ yields

$$d_X(y, y') \leq 2\sqrt{\tau} \| w \odot (y - y') \|_2 = 2\sqrt{\tau} \| W(y - y') \|_2.$$

Taking the supremum over $y, y' \in \mathcal{L}_M$ gives

$$\mathcal{D}(\mathcal{L}_M, d_X) \leq 2\sqrt{\tau} \cdot \mathrm{diam}(W\mathcal{L}_M, \| \cdot \|_2).$$

By (Vershynin, 2018, Proposition 7.5.4), it follows that

$$\mathrm{diam}(W\mathcal{L}_M, \| \cdot \|_2) \leq \sqrt{2\pi} \, \mathcal{W}(W\mathcal{L}_M).$$

Moreover, since $W\mathcal{L}_M = WM \, \mathcal{L}_{A'}$, the Gaussian width is monotone under linear maps and

$$\mathcal{W}(W\mathcal{L}_M) \leq \|WM\|_2 \, \mathcal{W}(\mathcal{L}_{A'}) =: \|M\|_{w,2} \, \mathcal{W}(\mathcal{L}_{A'}).$$

Combining the above inequalities, we obtain

$$\mathcal{D}(\mathcal{L}_M; d_X) \leq C\sqrt{\tau} \|M\|_{w,2} \, \mathcal{W}(\mathcal{L}_{A'}) \leq C\tau \mathcal{W}(\mathcal{L}_{A'}), \qquad C = 2\sqrt{2\pi}.$$

It remains to bound $\mathcal{W}(\mathcal{L}_{A'})$. Let $p = d + 1$ and set $K := B_2(A') \cap \frac{1}{\lambda} B_1$, so that $\mathcal{L}_{A'} \subseteq K$ and hence $\mathcal{W}(\mathcal{L}_{A'}) \leq \mathcal{W}(K)$. We have the bound $K \subseteq \frac{1}{\lambda} B_1$, which implies

$$\mathcal{W}(K) \leq \mathcal{W}\left( \frac{1}{\lambda} B_1 \right) = \frac{1}{\lambda} \mathcal{W}(B_1), \qquad \text{and} \qquad \mathcal{W}(B_1) = \mathbb{E} \max_{1 \leq j \leq p} |g_j| \leq C_0 \sqrt{\log(2p)}.$$

Thus, it follows that $\mathcal{W}(K) \leq O(\frac{\sqrt{\log(d)}}{\lambda})$. Plugging the bound on $\mathcal{W}(\mathcal{L}_{A'})$ into the upper bound of $\mathcal{D}(\mathcal{L}_M, d_X)$ yields the claim. $\qquad\square$

## B.2   BOUNDING THE METRIC ENTROPY

In this subsection, we establish an upper bound for the metric entropy $\mathcal{M}_{\mathcal{E}}$ of the space $\mathcal{L}_{w,M}$. To estimate this entropy, we first provide detailed definitions of covering numbers and metric entropy.

**Definition 14.** *Let $d_X$ be a pseudo-metric on $\mathbb{R}^d$. Given a vector $x \in \mathbb{R}^d$ and $t \geq 0$, we define the $d_X$-ball of radius $t$ centered at $x$ as $B_X(x,t) = \{x' \in \mathbb{R}^d \mid d_X(x,x') \leq t\}$.*

**Definition 15.** *Let $K, T \subseteq \mathbb{R}^d$ be two convex bodies. The covering number $N(K,T)$ represents the minimum number of copies of $T$ required to cover $K$*

$$N(K,T) = \min\{k \in \mathbb{N} : \exists\{x_i\}_{i=1}^k, K \subseteq \bigcup_{i=1}^k (x_i + T)\}.$$

*Let $d_X$ be a pseudo-metric and $t > 0$ be a scalar. The covering number of a set $K$ with respect to $d_X$ and radius $t$ is denoted by $N(K, d_X, t) = N(K, B_X(0,t))$, where $B_X(0,t)$ is the $d_X$-ball of radius $t$ centered at the origin. The metric entropy is given by $\mathcal{M}_{\mathcal{E}} = \log N(K, d_X, t)$.*

We now apply a standard tool, the Dual Sudakov minoration (Bourgain J & V., 1989), to bound the covering numbers in both the residual space and the $\ell_1$-penalty space. The following theorem provides an upper bound on the covering numbers of the Euclidean unit ball within a metric space using $\ell_p$-norm balls with radius $t > 0$.

**Definition 16.** *The Lévy mean of $\ell_p$ is defined as*

$$M_p = \frac{\mathbb{E}_{g \in \mathcal{N}(0, I_d)} \|g\|_p}{\mathbb{E}_{g \in \mathcal{N}(0, I_d)} \|g\|_2}.$$

**Theorem 17.** *(Dual Sudakov Minoration) Let $\|\cdot\|_p$ be a norm, and let $B_2 \subseteq \mathbb{R}^d$ denote the Euclidean unit ball, defined as $B_2 = \{x \in \mathbb{R}^d \mid \|x\|_2 \leq 1\}$. Then,*

$$\log N(B_2, \|\cdot\|_p, t) \leq O(d)\frac{M_p^2}{t^2}.$$

**Lemma 18.** *(Woodruff & Yasuda (2023), slightly modified) Let $q \geq 2$, let $M \in \mathbb{R}^{m \times (d+1)}$ be a matrix, and let $w \in \mathbb{R}^m$ be a weight vector. Then, for a standard Gaussian vector $g \sim \mathcal{N}(0, I)$, it holds that*

$$\mathbb{E}_{g \sim N(0, I_m)} [\|Mg\|_{w,q}] \leq m^{1/q} \cdot \sqrt{q\tau},$$

*and*

$$\mathbb{E}_{g \sim N(0, I_{d+1})} [\|g\|_2] \leq \sqrt{d+1}.$$

We now focus on the $\ell_1$-penalty space for the Gaussian process. To bound the metric entropy of the set $B_1(1/\lambda)$ using the unweighted $\ell_\infty$-ball, we decompose the process into two steps: covering the Euclidean unit ball with $B_\infty$, and covering $B_1$ using the Euclidean unit ball. We define the unweighted $\ell_p$ (including $\ell_\infty$) unit ball as $B_p = \{x \mid x \in \mathbb{R}^{d+1}, \|x\|_p \leq 1\}$. The following lemma provides a bound for the first step.

**Lemma 19.** *(Woodruff & Yasuda, 2023) Let $p \geq 2$ and let $B_p$ be the unit ball for the $\ell_p$ norm. Then,*

$$\log N(B_2, B_\infty, t) \leq O\left(\frac{\log d}{t^2}\right).$$

Since the $B_1$ ball has a non-smooth geometric structure, a substantial portion of its volume is concentrated near its center. This concentration implies that fewer smaller balls are needed to effectively cover the unit ball. A direct application of the $\epsilon$-net argument typically yields a general bound of $O((1 + \frac{1}{t})^{d+1})$ in the worst-case. To obtain the better bound by utilizing this concentration, we use the Sudakov minoration inequality (Vershynin, 2018), specifically for the non-smooth $B_1$ ball, as follows.

**Theorem 20.** *Let $\mathcal{K}$ be a convex body in $\mathbb{R}^d$, and let $N(\mathcal{K}, B_2, t)$ denote the covering number of balls of radius $t$ required to cover $\mathcal{K}$. Then, for any $t > 0$,*

$$\sqrt{\log N(\mathcal{K}, B_2, t)} \leq C \cdot \frac{\mathcal{W}(\mathcal{K})}{t},$$

*where $C$ is an absolute constant, and $\mathcal{W}(\mathcal{K}) = \mathbb{E}\left[\sup_{x \in \mathcal{K}} \langle g, x \rangle\right]$ represents the Gaussian width with respect to a standard Gaussian vector $g \sim \mathcal{N}(0, I_d)$.*

**Lemma 21.** *Let $B_1 = \{x \in \mathbb{R}^d \mid \|x\|_1 \leq 1\}$ and $B_\infty = \{x \in \mathbb{R}^d \mid \|x\|_\infty \leq 1\}$. For any $t \in (0,1]$, it holds that*

$$\log N(B_1, B_\infty, t) \leq O\left(\frac{\log d}{t}\right).$$

*Proof.* Fix $t \in (0,1]$ and let $\gamma \in [t,1]$. We use the standard chaining rule for covering numbers

$$\log N(B_1, B_\infty, t) \leq \log N(B_1, B_2, \gamma) + \log N(B_2, B_\infty, t/\gamma).$$

Let $g \sim \mathcal{N}(0, I_d)$. The Gaussian width of $B_1$ satisfies

$$\mathcal{W}(B_1) = \mathbb{E} \sup_{\|x\|_1 \leq 1} \langle g, x \rangle = \mathbb{E}\|g\|_\infty = \mathbb{E} \max_{1 \leq i \leq d} |g_i| \leq \sqrt{2 \log(2d)}.$$

By the Sudakov inequality (Theorem 20), there exists a universal constant $C_1$ such that

$$\log N(B_1, B_2, \gamma) \leq C_1 \frac{\mathcal{W}(B_1)^2}{\gamma^2} \leq C_1 \frac{2\log(2d)}{\gamma^2}.$$

Let $\|\cdot\|_X = \|\cdot\|_\infty$. By Definition 16, its Lévy mean is

$$M_X = \frac{\mathbb{E}\|g\|_\infty}{\mathbb{E}\|g\|_2} \leq \frac{\sqrt{2\log(2d)}}{c\sqrt{d}} \leq C_2 \sqrt{\frac{\log(2d)}{d}},$$

for universal constants $c, C_2 > 0$. Applying the dual Sudakov minoration theorem (for covering the Euclidean ball by $\|\cdot\|_\infty$-balls), we obtain that for $s \in (0,1]$,

$$\log N(B_2, B_\infty, s) \leq C_3 \, d \, \frac{M_X^2}{s^2} \leq C_4 \frac{\log(2d)}{s^2},$$

for universal constants $C_3, C_4$.

Substituting the two bounds into $\log N(B_1, B_\infty, t)$ yields

$$\log N(B_1, B_\infty, t) \leq C_5 \log(2d)\left(\frac{1}{\gamma^2} + \frac{\gamma^2}{t^2}\right).$$

Choose $\gamma = \sqrt{t} \in [t, 1]$ (since $t \in (0,1]$). Thus, we conclude that

$$\log N(B_1, B_\infty, t) \leq O\left(\frac{\log d}{t}\right),$$

which proves the claim. $\qquad\qquad\square$

**Lemma 22.** *(Munteanu & Omlor, 2024) Let $M \in \mathbb{R}^{m \times d}$ and let $w \in \mathbb{R}^m_{\geq 0}$ be a non-negative weight vector corresponding to the rows of $M$. Then, for any $1 \leq r \leq q$ and any $t > 0$,*

$$N(B_{1,r}(M), B_{1,q}(M), t) \geq N(B_{w,r}(M), B_{w,q}(M), t).$$

In the following, we give two upper bounds for the covering numbers in the $\ell_1$-penalty space based on the radius $t$. For larger radii ($t > t_0/\lambda$), the covering number scales as $(1/t)^2$, indicating a quadratic increase as $t$ decreases. Conversely, for smaller radii ($t \leq t_0/\lambda$), the covering number grows logarithmically with $1/t$.

**Lemma 23.** *Let $p = d + 1$, $\lambda > 0$, and let $M \in \mathbb{R}^{m \times p}$ have orthonormal columns. Define the set $B_\infty(M) = \{x \mid x \in \mathcal{T}', \|Mx\|_\infty \leq 1\}$ as the unit ball in the $\ell_\infty$-norm mapped by $M$. Let $H = \max_{1 \leq i \leq m} \|e_i^T M\|_\infty$, where $e_i \in \mathbb{R}^m$ is the $i$-th standard basis vector. Let $t_0 = O(H\sqrt{\frac{\log m}{d}})$. Then, the following bounds on the metric entropy hold for $t > t_0/\lambda$*

$$\log N(B_1(1/\lambda), B_\infty(M), t) \leq O(H^2)\frac{\log d \cdot \log m}{\lambda^2 t^2},$$

*and for $0 < t \leq t_0/\lambda$,*

$$\log N(B_1(1/\lambda), B_\infty(M), t) \leq O\left(d \log d/\lambda^2 + m \log\left(1 + \frac{t_0}{\lambda t}\right)\right).$$

*Proof.* Let $p = d + 1$, and assume that $M \in \mathbb{R}^{m \times p}$ has orthonormal columns, i.e., $M^\top M = I_p$. Let $\| \cdot \|_{H_m}$ be the norm $\|x\|_{H_m} := \|Mx\|_\infty$ on $\mathbb{R}^p$, and denote its unit ball by

$$H_m := \{x \in \mathbb{R}^p \mid \|x\|_{H_m} \leq 1\} = \{x \mid \|Mx\|_\infty \leq 1\}.$$

Note that $B_\infty(M) = H_m \cap \mathcal{T}'$, hence by monotonicity of covering numbers,

$$N(B_1(1/\lambda), B_\infty(M), t) \leq N\left(\frac{1}{\lambda}B_1, H_m, t\right) = N(B_1, H_m, \lambda t). \tag{1}$$

Let $H := \max_{1 \leq i \leq m} \|e_i^\top M\|_\infty$, and let $t_0 = O\left(H\sqrt{\frac{\log m}{d}}\right)$. By the Bernstein–Jackson (Carl-type) entropy bound for the embedding $(\mathbb{R}^p, \| \cdot \|_1) \to (\mathbb{R}^p, \| \cdot \|_{H_m})$ (see (Carl, 1985)), there exists a universal constant $C_1$ such that for all $s \geq t_0$,

$$\log N(B_1, H_m, s) \leq C_1 \frac{H^2 \log d \cdot \log m}{s^2}. \tag{2}$$

Substituting $s = \lambda t$ and using equation 1 yields, for $t \geq t_0/\lambda$,

$$\log N(B_1(1/\lambda), B_\infty(M), t) \leq \log N(B_1, H_m, \lambda t) \leq C_1 \frac{H^2 \log d \cdot \log m}{\lambda^2 t^2}.$$

For $0 < t \leq t_0/\lambda$, we use the standard volumetric bound in $(\mathbb{R}^p, \| \cdot \|_{H_m})$, leading to

$$\log N(B_1, H_m, \lambda t) \leq \log N(B_1, H_m, \lambda t_0) + \log N(t_0 H_m, H_m, \lambda t)$$
$$\leq C_1 \frac{H^2 \log d \cdot \log m}{\lambda^2 t^2} + m \log\left(1 + \frac{t_0}{\lambda t}\right)$$
$$\leq O\left(d \log d/\lambda^2 + m \log\left(1 + \frac{t_0}{\lambda t}\right)\right).$$

This completes the proof. $\qquad\square$

In the following lemma, we present two different upper bounds on the metric entropy of the intersection between the residual space $B_{w,2}(M)$ and the $\ell_1$-penalty space $B_1(1/\lambda)$. Specifically, we employ the weighted $\ell_\infty$ unit ball to cover both the weighted ball $B_{w,2}(M)$ and the unweighted ball $B_1$ with the same radius. We then provide bounds for two cases: when the radius $t$ is larger than $t_0$, and when $t$ is smaller than $t_0$. Let $\tau = \sup_{x' \in \mathcal{L}_{w,M}} \|Mx'\|_{w,2,\infty}^2$ be the weighted maximum of $\ell_2$ leverage score, and define $G = 1 + \mathcal{E} = 1 + \sup_{x' \in \mathcal{L}} \left| \|SA'x'\|_{w,2}^2 - \|A'x'\|_2^2 \right|$.

**Lemma 24.** *Let $\lambda > 0$, and let $\mathcal{L}_{w,M} = \{y = Mx \mid x \in B_{w,2}(M) \cap B_1(1/\lambda)\}$. For any $t \in (0, 1]$, the metric entropy of $\mathcal{L}_{w,M}$ with respect to the metric $d_X$ satisfies the following bounds, for $0 < t < t_0/\lambda$,*

$$\log N(\mathcal{L}_{w,M}, d_X, t) \leq \min\{O(d \log \frac{Gm}{t}), O(d \log d/\lambda^2 + m \log(1 + \frac{Gt_0}{\lambda t}))\},$$

*and for $t > t_0/\lambda$,*

$$\log N(\mathcal{L}_{w,M}, d_X, t) \leq O(\frac{\tau G^2 \log m}{t^2} \cdot \min\{1, \frac{\log d}{\lambda^2}\}),$$

*where $t_0 = \tau\sqrt{\frac{\log m}{d}}$.*

*Proof.* For all $y, y' \in B_{w,2}(M)$, we have $d_X(y, y') \leq 2\|y - y'\|_{w,\infty,2}$ (hence, in particular, $d_X$ is controlled by a weighted $\ell_\infty$-type norm after the embedding used below). Next, we define the matrix $M_w \in \mathbb{R}^{m \times (d+1)}$ such that each row of $M_w$ is obtained by multiplying the corresponding entry of the weight vector $w$ by the respective row of the matrix $M$, i.e., $(M_w)_i = \sqrt{w_i} \cdot M_i$. Since $w_i = 1/p_i$ represents the weight of the $i$-th row and $p_i$ is the sampling probability, we have $w_i \geq 1$ for all $i$. Then, the convex body $B_{w,2}(M, )$ is equal to $B_2(M_w, G)$, since

$$B_{w,2}(M) = \left\{y \in \text{range}(M) \mid \sum_{i=1}^{m} w_i y_i^2 \leq 1\right\} = B_2(M_w).$$

Thus, for any $t > 0$, we have

$$\log N(B_{w,2}(M, G), d_X, t/G) \leq \log N(B_2(M_w), 2\|M_w \cdot \|_{w,\infty}, t/G)$$
$$= \log N\left(B_2(M_w), B_\infty(M_w), \frac{t}{2G}\right).$$

By Lemma Lemma 4 and Lemma Lemma 22 in Appendix, the following inequality holds

$$\log N(B_{w,2}(M), d_X, t) \leq O(d \log \frac{Gm}{t}).$$

Moreover, by a slight adaptation to Lemma 4, we also have $\log N(B_{w,2}(M), d_X, t) \leq O(\log m \frac{G^2 \tau}{t^2})$.

Let $H = \max_{1 \leq i \leq m} \|e_i^T M\|_\infty$ be the maximum row-wise $\ell_\infty$-norm of matrix $M$. By the inequality $\|x\|_\infty \leq \|x\|_2$ for any vector $x$, it follows that $H = \max_{1 \leq i \leq m} \|e_i^T M\|_\infty \leq \max_{1 \leq i \leq m} \|e_i^T M\|_2 \leq \sqrt{\tau}$. Consequently, applying Lemma 23 in Appendix, we obtain the following bounds on the metric entropy $\log N(B_1(1/\lambda), B_\infty(M_w), t/2)) \leq O(\tau \frac{\log d \cdot \log m}{\lambda^2 t^2})$ for $t > t_0/\lambda$, and the inequality $\log N(B_1(1/\lambda), B_\infty(M_w), t/2)) \leq O\left(d \log d/\lambda^2 + m \log\left(1 + \frac{t_0}{\lambda t}\right)\right)$ for $0 < t \leq t_0/\lambda$, where $t_0 = O(\tau \sqrt{\frac{\log m}{d}})$.

Next, we consider the metric entropy on the $\mathcal{L}_{w,M}$

$$\log N(\mathcal{L}_{w,M}, d_X, t) \leq \log N(\mathcal{L}_{w,M}, 2\|M_w \cdot \|_\infty, \frac{t}{G})$$
$$= \log N(B_{w,2}(M) \cap B_1(1/\lambda), 2\|M_w \cdot \|_\infty, \frac{t}{G})$$
$$\leq \min\{\log N(B_{w,2}(M), 2\|M_w \cdot \|_\infty, \frac{t}{2G}), \log N(B_1(1/\lambda), 2\|M_w \cdot \|_\infty, \frac{t}{2G})\}.$$

Combining the above inequalities, we conclude that

$$\log N(\mathcal{L}_{w,M}, d_X, t) \leq \min\{O(d \log \frac{Gm}{t}), O(d \log d/\lambda^2 + m \log(1 + \frac{Gt_0}{\lambda t}))\},$$

for $0 < t < t_0/\lambda$, and

$$\log N(\mathcal{L}_{w,M}, d_X, t) \leq O(\tau G^2) \min\{O(\frac{\log m}{t^2}), \frac{\log d \cdot \log m}{\lambda^2 t^2}\},$$

for $t > t_0/\lambda$. $\qquad\square$

### B.3 COMPUTING THE ENTROPY INTEGRAL

In this subsection, we bound the entropy integral of these $t$-nets using the following Dudley inequality (Vershynin, 2018) for Gaussian processes.

**Theorem 25.** *(Dudley inequality,(Vershynin, 2018)) Let $(X(t))_{t \in T}$ be a standard Gaussian process defined on a measurable space with a pseudo-metric $d_X$. Then, it holds that*

$$\mathbb{E}\left[\sup_{t \in T} X_t\right] \leq C \int_0^\infty \sqrt{\log N(T, d_X, t)} \, dt,$$

*where $T$ is a convex set, $C$ is an absolute constant, and $X_t$ alue of the Gaussian process indexed by $t \in T$.*

**Lemma 26.** *(Woodruff & Yasuda, 2023) Let $0 < \delta \leq 1$ and $C$ be a positive constant. Then,*

$$\int_0^\delta \sqrt{\log \frac{C}{t}} \, dt \leq \delta \left(\sqrt{\log \frac{C}{\delta}} + \frac{C\sqrt{\pi}}{2}\right).$$

**Lemma 27.** *Let $M \in \mathbb{R}^{m \times (d+1)}$ be a matrix, where $m = \tilde{O}(d + \log(1/\delta))$, and let $\lambda, \delta > 0$. Then, the metric entropy of $\mathcal{L}_{w,M}$ satisfies*

$$\int_0^\infty \sqrt{\log N(\mathcal{L}_{w,M}, d_X, t)}\, dt \leq O\left(G\sqrt{\tau \log m}\right)\left(1 + \log d \cdot \min\left\{1, \frac{\sqrt{\log d}}{\lambda}\right\}\right),$$

*where $\tau$ is the maximum weighted $\ell_2$-leverage score of $M$.*

*Proof.* Note that it suffices to integrate the entropy integral from $0$ to the diameter $\mathcal{D} = \mathrm{diam}(\mathcal{L}_{w,M})$, because for $t > \mathcal{D}$, the entropy is zero. Let $t_0 = O(\tau\sqrt{\frac{\log m}{d}})$, and let $t'$ be a radius with $t' \in [t_0/\lambda, \mathcal{D}]$. For small radii $t < t'$, we use the first bound of Lemma 24 in Appendix as follows

$$\log N(\mathcal{L}_{w,M}, d_X, t) \leq \min\{O(d\log \frac{Gm}{t}), O(d\log d/\lambda^2 + m\log(1 + \frac{Gt_0}{\lambda t}))\}.$$

By Lemma 26 in Appendix, the entropy integral is bounded by

$$\int_0^{t'} \sqrt{\log N(\mathcal{L}_{w,M}, d_X, t)}\, dt$$

$$\leq \min\{\int_0^{t'} \sqrt{O(d\log \frac{Gm}{t})}\, dt, \int_0^{t'} \sqrt{O(m\log(1 + \frac{Gt_0}{t\lambda}) + m\log m/\lambda^2)}\, dt\}$$

$$\leq O(t') \cdot \min\left\{\sqrt{d\log \frac{Gm}{t'}}, \sqrt{m}\sqrt{\log\left(1 + \frac{Gt_0}{\lambda t'}\right)} + \frac{\sqrt{m\log m}}{\lambda}\right\}$$

$$\leq O(t') \cdot \min\left\{\sqrt{d\log \frac{Gm}{t'}}, \sqrt{m\log\left(1 + \frac{G\tau}{\lambda t'}\sqrt{\frac{\log m}{d}}\right)} + \frac{\sqrt{m\log m}}{\lambda}\right\}$$

On the other hand, for large radii $t > t'$, we use the second bound of Lemma 24 (in Appendix), which gives

$$\log N(\mathcal{L}_{w,M}, d_X, t) \leq O(\frac{\tau G^2 \log m}{t^2} \cdot \min\{1, \frac{\log d}{\lambda^2}\}).$$

Combining these inequalities, we obtain

$$\int_{t'}^{\mathcal{D}} \sqrt{\log N(\mathcal{L}_{w,M}, d_X, t)}\, dt \leq O(1)\sqrt{\tau G^2 \log m} \cdot \min\{1, \frac{\sqrt{\log d}}{\lambda}\} \int_{t'}^{\mathcal{D}} \frac{1}{t}\, dt$$

$$= O(1)\sqrt{\tau G^2 \log m} \cdot \min\{1, \frac{\sqrt{\log d}}{\lambda}\} \log\left(\frac{G}{t'}\right).$$

By setting $m = \tilde{O}(d + \log(1/\delta))$, applying Lemma 3, and choosing the radius $t' = G\sqrt{\tau/d}$, we obtain

$$\int_0^\infty \sqrt{\log N(\mathcal{L}_{w,M}, d_X, t)}\, dt$$

$$\leq \int_0^{t'} \sqrt{\log N(\mathcal{L}_{w,M}, d_X, t)}\, dt + \int_{t'}^{\mathcal{D}} \sqrt{\log N(\mathcal{L}_{w,M}, d_X, t)}\, dt$$

$$\leq O(G\sqrt{\tau}) \cdot \min\left\{\sqrt{\log m}, \sqrt{\frac{m}{d}}\sqrt{\log\left(1 + \frac{\sqrt{\log m}}{\lambda}\right)} + \frac{\sqrt{m\log m}}{\lambda\sqrt{d}}\right\}$$

$$+ O\left(G\sqrt{\tau\log m}\right) \cdot \min\left\{1, \frac{\sqrt{\log d}}{\lambda}\right\} \cdot \log d$$

$$\leq O\left(G\sqrt{\tau\log m}\right)\left(1 + \log d \cdot \min\left\{1, \frac{\sqrt{\log d}}{\lambda}\right\}\right).$$

$\square$

**Lemma 28.** *(Woodruff & Yasuda, 2023) Let $A' \in \mathbb{R}^{n \times (d+1)}$ and $\lambda > 0$. Let $\Lambda = \sup_{\|A'x\|_2^2 + \lambda\|x\|_1 \leq 1, x \in \mathbb{R}^{d+1}} \left| \sum_{i=1}^n g_i |[A'x](i)|^2 \right|$. Given a convex set $L$, let $\mathcal{M}_\varepsilon$ be the metric entropy of $L$, and let $\mathcal{D}$ Gaussian complexity associated with $L$. Then,*

$$\mathbb{E}_{g \sim \mathcal{N}(0, I_n)}[|\Lambda|^l] \leq (2\mathcal{M}_\varepsilon)^l(\mathcal{M}_\varepsilon/\mathcal{D}) + O(\sqrt{l}\mathcal{D})^l.$$

**Lemma 29.** *Let $A' \in \mathbb{R}^{n \times (d+1)}$ and let $S \in \mathbb{R}^{n \times n}$ be a diagonal sampling matrix with exactly $m$ nonzero diagonal entries. Assume that with probability at least $3/4$,*

$$\|SA'x\|_{w,2}^2 = (1 \pm \tfrac{1}{2})\|A'x\|_2^2 \qquad \text{for all } x \in \mathbb{R}^{d+1}. \tag{3}$$

*Let $I := \{i \in [n] \mid S_{ii} \neq 0\}$ be the sampled index set, and define*

$$\mathcal{G}(SA') := \sum_{i \in I} \sup_{SA'x \neq 0} \frac{w_i|(SA')_{i:}x|^2 + \frac{\lambda}{n}\|x\|_1}{\|SA'x\|_{w,2}^2 + \lambda\|x\|_1}.$$

*Then*

$$\Pr\left[\mathcal{G}(SA') \leq 8\,\mathcal{G}(A')\right] \geq \frac{1}{2}.$$

*Proof.* Define for $i \in [n]$ the (unweighted) sensitivity

$$\varrho_i(A') := \sup_{A'x \neq 0} \frac{|A'_{i:}x|^2 + \frac{\lambda}{n}\|x\|_1}{\|A'x\|_2^2 + \lambda\|x\|_1},$$

and define the distortion factor

$$\Gamma(S) := \sup_{SA'x \neq 0} \frac{\|A'x\|_2^2 + \lambda\|x\|_1}{\|SA'x\|_{w,2}^2 + \lambda\|x\|_1}.$$

On the event equation 3, we have $\|SA'x\|_{w,2}^2 \geq \frac{1}{2}\|A'x\|_2^2$ for all $x$, hence for all $x$,

$$\frac{\|A'x\|_2^2 + \lambda\|x\|_1}{\|SA'x\|_{w,2}^2 + \lambda\|x\|_1} \leq \frac{\|A'x\|_2^2 + \lambda\|x\|_1}{\frac{1}{2}\|A'x\|_2^2 + \lambda\|x\|_1} \leq 2,$$

and therefore $\Gamma(S) \leq 2$. Consequently, we get

$$\Pr\left[\Gamma(S) \leq 2\right] \geq \frac{3}{4}.$$

Fix $i \in I = \{i \mid i \in [n], S_{ii} \neq 0\}$. Since $S$ is diagonal, $(SA')_{i:} = S_{ii}A'_{i:}$ and $S_{ii} \neq 0$. In particular, $|(SA')_{i:}x|^2 = S_{ii}^2|A'_{i:}x|^2$. Since $w_i \geq 1$, we have

$$w_i|(SA')_{i:}x|^2 + \frac{\lambda}{n}\|x\|_1 \leq w_i S_{ii}^2\left(|A'_{i:}x|^2 + \frac{\lambda}{n}\|x\|_1\right).$$

Thus, for every $i \in I$, we get

$$\sup_{SA'x \neq 0} \frac{w_i|(SA')_{i:}x|^2 + \frac{\lambda}{n}\|x\|_1}{\|SA'x\|_{w,2}^2 + \lambda\|x\|_1} \leq w_i S_{ii}^2 \sup_{SA'x \neq 0} \frac{|A'_{i:}x|^2 + \frac{\lambda}{n}\|x\|_1}{\|A'x\|_2^2 + \lambda\|x\|_1} \cdot \sup_{SA'x \neq 0} \frac{\|A'x\|_2^2 + \lambda\|x\|_1}{\|SA'x\|_{w,2}^2 + \lambda\|x\|_1}$$

$$\leq w_i S_{ii}^2 \varrho_i(A') \cdot \Gamma(S).$$

Summing over $i \in I$ gives

$$\mathcal{G}(SA') \leq \Gamma(S) \sum_{i \in I} w_i S_{ii}^2 \varrho_i(A') \leq \Gamma(S) \sum_{i=1}^n w_i S_{ii}^2 \varrho_i(A').$$

Then, we have

$$\mathbb{E}\left[\sum_{i=1}^n w_i S_{ii}^2 \varrho_i(A')\right] = \sum_{i=1}^n \mathbb{E}[w_i S_{ii}^2]\varrho_i(A') = \sum_{i=1}^n \varrho_i(A') = \mathcal{G}(A').$$

By Markov's inequality,

$$\Pr\left[\sum_{i=1}^n w_i S_{ii}^2 \varrho_i(A') \leq 4\mathcal{G}(A')\right] \geq \frac{3}{4}.$$

With probability at least $\frac{1}{2}$, both events $\{\Gamma(S) \leq 2\}$ and $\{\sum_i w_i S_{ii}^2 \varrho_i(A') \leq 4\mathcal{G}(A')\}$ hold. Therefore, it follows that $\Pr[\mathcal{G}(SA') \leq 8\mathcal{G}(A')] \geq \frac{1}{2}$. $\qquad\square$

In the following theorem, we present the main result provides a bound on the $l$-th moment of the sampling error $\mathbb{E}|\mathcal{E}|^l$.

**Theorem 30.** *Let $A' \in \mathbb{R}^{n \times (d+1)}$ be an input matrix, let $S$ be a random sampling matrix, and let $\varepsilon, \delta \in (0,1)$ and $\lambda > 0$ be parameters. If $\alpha = \tilde{O}\left(\frac{1}{\epsilon^2} \cdot \left(\log\left(d\log(\delta^{-1})\right)(\ln d)^2 \cdot \min\left\{1, \frac{\log d}{\lambda^2}\right\} + \log\frac{1}{\delta}\right)\right)$ and for all $i \in [n]$ it holds that*

$$p_i \geq \min\{1, \alpha(\tau_{i,2}(A') + \frac{1}{n})\},$$

*where $\tau_{i,2}(A')$ denotes the $\ell_2$ leverage score of the $i$-th row of $A'$. Then, with failure probability at most $\delta$, it holds that for all $x \in \mathbb{R}^{d+1}$ with $x_{d+1} = 1$,*

$$\|SA'x\|_{w,2}^2 + \lambda\|x\|_1 \leq (1 \pm \epsilon)(\|A'x\|_2^2 + \lambda\|x\|_1),$$

*and the coreset size is at most*

$$m = \tilde{O}\left(\frac{d\log^3 d}{\epsilon^2} \cdot \min\{1, \frac{\log d}{\lambda^2}\} + \frac{d}{\epsilon^2}\log\frac{1}{\delta}\right).$$

*Proof.* By the construction of the sampling matrix $S$, for any $i \in [n]$, the sampling probability satisfies $0 < p_i \leq 1$, and the corresponding sampling weight is $S_{ii} = 1/p_i \geq 1$. This implies that $\mathbb{E}(\|SA'x\|_{w,2}^2 + \lambda\|x\|_1) = \|A'x\|_2^2 + \lambda\|x\|_1$ for $\lambda > 0$ and any vector $x$. We set $\alpha = \tilde{O}\left(\frac{1}{\epsilon^2} \cdot \log^3 d \min\{1, \frac{\log d}{\lambda^2}\} + \log\frac{1}{\delta}\right)$, where $\sqrt{l} = \tilde{O}\left(\frac{\lambda}{\epsilon}\log^{5/2} d\log\frac{1}{\delta}\min\left\{1, \frac{\log d}{\lambda^2}\right\}\right)$ denotes the maximum number of finite moments of the sampling error $\mathcal{E}$. To bound the coreset size $m$, let $X_i$ be the indicator random variable that represents whether the $i$-th row is included in $S$. Applying Lemma 10 in Appendix, we get

$$\mathbb{E}\left(\sum_{i=1}^n X_i\right) = \sum_{i=1}^n p_i = \alpha\left(1 + \sum_{i=1}^n (2\tau_i + \frac{1}{n})\right) = \alpha(2 + 2d) \leq 4\alpha d.$$

Similarly, we can derive the lower bound of $m$ as follows

$$\mathbb{E}\left(\sum_{i=1}^n X_i\right) = \sum_{i=1}^n p_i \geq \alpha\left(\sum_{i=1}^n 2\tau_i\right) = 2\alpha d.$$

By applying the Chernoff inequality, we have

$$m = \sum_{i=1}^n X_i \leq 2 \cdot \mathbb{E}\left(\sum_{i=1}^n X_i\right) \leq 8\alpha d$$

with failure probability at most $2\exp\left(-\mathbb{E}(\sum_{i=1}^n X_i)/3\right) \leq 2\exp(-\frac{2\alpha d}{3}) \leq \delta$.

Applying Lemma 1, the analysis of the empirical process associated with the sampling error can be reduced to a Gaussian process. Specifically, we obtain

$$\mathbb{E}_S \sup_{\|A'x\|_2^2 + \lambda\|x\|_1 = 1, x \in \mathcal{T}'} |\|SA'x\|_{w,2}^2 - \|A'x\|_2^2|^l$$

$$\leq (2\pi)^{l/2} \mathbb{E}_{S,g} \sup_{\|A'x\|_2^2 + \lambda\|x\|_1 = 1, x \in \mathcal{T}'} \left|\sum_{i \in Q} g_i w_i |A'_i x|^2\right|^l,$$

where $l > 1$ is an integer, $w_i$ denotes the weight for sampling the $i$-row, $Q$ the indices of nonzero diagonal entries in $S$, and $g \sim \mathcal{N}(0, I_m)$ is a standard Gaussian vector.

Next, we define $\Lambda = \sup_{\|A'x\|_2^2 + \lambda\|x\|_1 = 1, x \in \mathcal{T}'}\left(\sum_{i \in S} g_i w_i |A'_i x|^2\right)$ for the random sampling matrix $S$. To further bound the quantity $\Lambda$, we apply Lemma 28, which connects $\Lambda$ with the metric entropy $\mathcal{M}_{\mathcal{E}}$ and the diameter $\mathcal{D}$ of geometric body resulted by the Gaussian process. This gives us the following bound

$$\mathbb{E}_{g \sim \mathcal{N}(0, I_m)}[|\Lambda|^l] \leq (2\mathcal{M}_{\mathcal{E}})^l(\mathcal{M}_{\mathcal{E}}/\mathcal{D}) + O(\sqrt{l}\mathcal{D})^l$$

for a fixed $l$.

Let $M = SA'$ denote the sampled and rescaled matrix with $m$ rows, and let $w$ represent the weight vector corresponding to each row of $M$. Next, we bound the maximum weight leverage score $\tau = \sup_{\|Mx\|_{w,2}=1, i \in [m]} w_i |M_i x|^2$.

We set the number of samples $m$ to be $\tilde{O}(d + \log(1/\delta))$ using the $\ell_2$ leverage scores sampling method (Cohen et al., 2015), which achieves $\|S'A'x\|_{w,2}^2 \leq (1 \pm \frac{1}{2})\|A'x\|_2^2$ for a fixed sampling matrix $S'$ with probability at least $1 - \delta$. By applying Lemma 29 in Appendix and the definition of sampling probability, we have $\tau \leq O(1/\alpha)$.

According to Lemma 6, by choosing the constants in $\alpha$ sufficiently large, we obtain a bound on the metric entropy

$$O\left(G\sqrt{\tau \log m}\right)\left(1 + \log d \cdot \min\left\{1, \frac{\sqrt{\log d}}{\lambda}\right\}\right)$$
$$\leq O\left(G\alpha^{-1/2}\sqrt{\log m}\right)\left(1 + \log d \cdot \min\left\{1, \frac{\sqrt{\log d}}{\lambda}\right\}\right)$$
$$\leq G\epsilon/8 := \mathcal{M}_{\mathcal{E}}.$$

By Lemma 3, we derive a bound on the diameter $O(\tau\sqrt{\log d}/\lambda) \leq O(\alpha^{-1}\sqrt{\log d}/\lambda) \leq \frac{\epsilon}{2\sqrt{l}} := \mathcal{D}$.

By combining the bounds on the metric entropy $\mathcal{M}_{\mathcal{E}}$ and the diameter $\mathcal{D}$, we ensure the sampling error $\mathbb{E}_{g \sim \mathcal{N}(0,I_m)}[|\Lambda|^l] \leq \epsilon^l \delta$. Since the sampling error $\mathcal{E} = \sup_{x' \in \mathcal{L}_M} \left|\|SA'x'\|_{w,2}^2 - \|A'x'\|_2^2\right|$, we have $\mathcal{E}^l \leq 3^l \varepsilon^l \delta$, which yields $\mathbb{E}|\mathcal{E}|^l \leq (3\varepsilon)^l \delta$. By using Markov's inequality, we have $\mathcal{E} \leq 3\varepsilon$ with probability at least $1 - \delta$. $\qquad\square$

### B.4 Omitted Proofs of Lower Bound for Coreset Size

In this section, we provide the lower bound of the coreset size for LASSO regression, using a standard information-theoretic approach (Wang et al., 2010; Wainwright, 2009; Parulekar et al., 2021; Mai et al., 2023) based on Fano's inequality and KL divergence computations. Here, we start by constructing a hard instance over the $k$-sparse supports.

Let $\mathcal{C} \subset \{0,1\}^d$ be a set of $k$-sparse binary vectors (i.e., each vector has exactly $k$ nonzero entries), such that $|\mathcal{C}| = N \geq (d/k)^k$ and any two distinct vectors $c^{(i)}, c^{(j)} \in \mathcal{C}$ satisfy $|supp(c^{(i)}) \cap supp(c^{(j)})| \leq Ck$ for some constant $C \in (0,1)$. Such a codebook can be constructed using standard techniques from coding theory. For each codeword $c^{(i)} \in \mathcal{C}$, we define

$$v^{(i)} = \left[1, \frac{\epsilon}{\sqrt{k}} c^{(i)}\right] \in \mathbb{R}^{d+1}.$$

Let $G \in \mathbb{R}^{m \times d}$ be a matrix with i.i.d. standard Gaussian entries. Define the data matrix $Z^i = G(I + v^{(i)}v^{(i)\top})^{1/2}$. Then, each row $z_j^i \sim \mathcal{N}(0, I + v^{(i)}v^{(i)\top})$, and the data distribution is

$$\mathbb{P}_i = \mathcal{N}(0, I + v^{(i)}v^{(i)\top}).$$

We show that exact support recovery is impossible with fewer measurements than those suggested by the information-theoretic lower bound, given the input distribution.

**Lemma 31.** *Let $\epsilon \in (0,1)$, and let $v \in \mathbb{R}^{d+1}$ be the vector with $v = (1, \frac{1}{\sqrt{k}}c)$, where $c$ is a codeword uniformly chosen from $\mathcal{C}$. Let $P_i$ be the multivariate Gaussian distribution with covariance $I + vv^\top$. Then, for any estimator attempting to recover a $k$-sparse vector $c$, with at least $1/2$ probability, the number of samples $m$ must satisfy*

$$m \geq \Omega\left(\frac{k \log(d/k)}{\epsilon^2}\right).$$

*Proof.* Let $\mathbb{P}_i, \ldots, \mathbb{P}_N$ be the distributions constructed above. By Fano's inequality, we have

$$\Pr[error] \geq 1 - \frac{\frac{1}{N^2}\sum_{i \neq j} D_{\mathrm{KL}}(\mathbb{P}_i \| \mathbb{P}_j) + \log 2}{\log(N-1)},$$

where $D_{\mathrm{KL}}(\mathbb{P}_i||\mathbb{P}_j)$ denotes the Kullback-Leibler divergence between distributions $\mathbb{P}_i$ and $\mathbb{P}_j$.

To ensure the error probability is less than $1/2$, it suffices to ensure

$$\frac{1}{N^2} \sum_{i \neq j} D_{\mathrm{KL}}(\mathbb{P}_i||\mathbb{P}_j) \leq \frac{1}{4} \log N.$$

According to the definition of $\mathbb{P}_i$, the KL divergence between two such distributions is

$$D_{\mathrm{KL}}(\mathbb{P}_i||\mathbb{P}_j) = m \cdot D_{\mathrm{KL}}(\mathcal{N}(0, \Sigma_i)||\mathcal{N}(0, \Sigma_j)),$$

where $\Sigma_i = I + v^{(i)}v^{(i)\top}$. Using the formula for KL divergence between zero-mean Gaussian distribution, we have

$$D_{\mathrm{KL}}(\mathcal{N}(0, \Sigma_i)||\mathcal{N}(0, \Sigma_j)) = \frac{1}{2}(tr(\Sigma_j^{-1}\Sigma_i) - d + \log \frac{\det \Sigma_j}{\det \Sigma_i})$$

Since $\Sigma_i$ is a rank-1 perturbation, by the Sherman-Morrison formula, we have $\det(\Sigma_i) = 1 + \|v^{(i)}\|_2^2$ and $\Sigma_i^{-1} = I - \frac{v^{(i)}v^{(i)\top}}{1 + \|v^{(i)}\|_2^2}$. Thus, we obtain

$$D_{\mathrm{KL}}(\mathbb{P}_i||\mathbb{P}_j) = \frac{n}{2}(\|v^{(i)}\|_2^2 - \frac{\|v^{(j)}\|_2^2 + (v^{(j)\top}v^{(i)})^2}{1 + \|v^{(i)}\|_2^2} + \log \frac{1 + \|v^{(j)}\|_2^2}{1 + \|v^{(i)}\|_2^2})$$

For any $i \in [N]$, it holds that $\|v^{(i)}\|_2^2 = 1 + \epsilon^2$. Similarity, for any $i \neq j$, the inner product satisfies

$$(v^{(j)\top}v^{(i)})^2 = \left(1 + \frac{\epsilon^2}{k}\langle c^{(i)}, c^{(j)}\rangle\right)^2 \leq (1 + C\epsilon^2)^2.$$

Plugging these bounds into the expression for KL divergence, we get

$$D_{\mathrm{KL}}(\mathbb{P}_i||\mathbb{P}_j) \leq O(\epsilon^2 m).$$

Let $\log N = \Theta(k \log(d/k))$. Applying Fano's inequality, it holds that

$$\Pr[error] \geq 1 - \frac{mC\epsilon^2 + \log 2}{\log N}.$$

To ensure $\Pr[error] \leq 1/2$, we have

$$C\epsilon^2 \cdot m \geq \frac{1}{2}\log N = \Omega(k \log(d/k)) \rightarrow m \geq \Omega(\frac{k \log(d/k)}{\epsilon^2}).$$

$\square$

Our proof method, while differing in approach from previous work (Wainwright, 2009; Mai et al., 2023) that focuses on sketching algorithms, is based on similar ideas. In particular, by analyzing the coreset algorithm on a constructed hard instance, we establish a lower bound on the sample size required by any algorithm to achieve a $(1 + \epsilon)$-approximation on this constructed hard instance.

**Lemma 32.** *Let $A \in \mathbb{R}^{n \times d}$, $b \in \mathbb{R}^n$, and $\lambda \in (0, 1)$. Assume that $\|A\|_2 \leq 1$ and $\|b\|_2 \leq 1$. Let $S$ be a diagonal sampling matrix with $m$ nonzero entries. Suppose there exists an estimator that returns $\tilde{x} = \arg\min_{x \in \mathbb{R}^d} \|SAx - Sb\|_2^2 + \lambda\|x\|_1$ and satisfies*

$$\|A\tilde{x} - b\|_2^2 + \lambda\|\tilde{x}\|_1 \leq (1 + \epsilon) \cdot \min_{x \in \mathbb{R}^d}(\|Ax - b\|_2^2 + \lambda\|x\|_1).$$

*Then, the coreset size $m$ must satisfy*

$$m = \begin{cases} \Omega(\frac{\log d}{\lambda^2 \cdot \epsilon^2}), & \text{if } \lambda = \Omega(\frac{1}{\sqrt{d}}) \\ \Omega(\frac{d}{\epsilon^2} \log d), & \text{if } \lambda = O(\frac{1}{\sqrt{d}}) \end{cases}.$$

*Proof.* We prove this result using a similar approach to Theorem 13 in (Mai et al., 2023). We take $[b \ A] \sim \frac{1}{\sqrt{n}} G(I + vv^\top)^{1/2}$, where $v$ is a codeword in the set $\mathcal{C}$. Let $S$ be a sampling matrix that selects $m$ rows of $A$ and $b$. Since only the row indices selected by $S$ affect the coreset, and the weights can be absorbed into the analysis via rescaling, we may, without loss of generality, assume that all nonzero diagonal entries of $S$ are equal to 1. Under this assumption, the compressed matrix $SG(I + vv^\top)^{1/2}$ has the same distribution as $G(I + vv^\top)^{1/2}$.

By the concentration properties of Gaussian matrices (see Exercise 4.7.3 in (Vershynin, 2018)), with high probability, the LASSO objective satisfies

$$\|Ax - b\|_2^2 + \lambda\|x\|_1 \approx 1 + \|x\|_2^2 + (1 - \epsilon c^T x)^2 + \lambda\|x\|_1 =: L(x),$$

where $v = (1 \ c)$. Since $L(x)$ is a 1-strongly convex function, we get

$$L(\hat{x}) \geq L(x^*) + \|\hat{x} - x^*\|_2^2.$$

for any $\hat{x}$, where $x^*$ is the minimizer of $L(x)$.

Fix $\epsilon = 1/2$, we set $\lambda = \frac{1}{2\sqrt{k}}$. Here, it holds that $x^* = c/5$ and $L(x^*) \approx 2$. Suppose there exist a estimator algorithm satisfies $L(\hat{x}) \leq (1 + c_1)L(x^*)$ for a sufficiently small $c_1$. Then we have

$$(1 + c_1)L(x^*) \geq L(x^*) + \|\hat{x} - x^*\|_2^2,$$

which means the gap $\|\hat{x} - x^*\|_2^2 \leq 2 \cdot c_1$.

Choosing a small enough constant $c_1$, we can recover $\mathrm{supp}(v)$. By Lemma 31, if $\frac{1}{4\lambda^2} = o(d)$, the required lower bound of $\Omega(\frac{1}{\lambda^2\epsilon^2} \log d)$ on the coreset size; if $\frac{1}{4\lambda^2} = \Theta(d)$, the required size is at least $\Omega(\frac{d}{\epsilon^2} \log d)$. □

## C   COMPLEMENTARY EXPERIMENTS

### C.1   EXPERIMENTS ON SKETCHING ALGORITHMS AND LASSO-SENS

In this section, we present experimental results comparing the performance of our proposed LASSO-Sens algorithm with a sketching-based algorithm for solving LASSO regression. We also acknowledge recent advances in sketching for LASSO, such as the work in Mai et al. (2023), which utilizes random projections to accelerate the optimization process.

To ensure a fair comparison, we follow the same experimental setup used in Section 4, conducting experiments on the Synthetic, CTs, and mediamill datasets with identical coreset sizes and regularization parameters. We evaluate algorithm performance in terms of loss, runtime, and sparsity. For the sketching-based algorithm, we set the number of sketching rows equal to the coreset size and run each experiment 10 times, reporting the average results.

As shown in Tables 6, and 7, 8, the proposed LASSO-Sens algorithm consistently achieves lower loss values than the sketching method on both small- and large-scale datasets. On the large-scale mnist8m dataset, LASSO-Sens is up to 10 times faster than the sketching algorithm when the coreset size is set to $m = \{15, 20\} \times d$. Moreover, on the Synthetic dataset, the sparsity of the LASSO-Sens solution is much lower than that of the sketching algorithm. Overall, the experimental results show that the proposed algorithm achieves lower regression loss and lower sparsity, particularly on large-scale datasets.

### C.2   EXPERIMENTS ON SENSITIVITY SAMPLING FOR STANDARD AND MODIFIED LASSO OBJECTIVES

In this section, we compare the performance of the sensitivity sampling algorithm on both the standard LASSO objective and the modified LASSO objective proposed in Chhaya et al. (2020), which takes the form $\|Ax - b\|_2^2 + \lambda\|x\|_1^2$. In Section 4, we used the FISTA algorithm to solve the standard LASSO problem, as it leverages the proximal operator of the $\ell_1$ norm. However, this solver is not applicable to the modified LASSO formulation, which involves a squared $\ell_1$ regularization term and lacks an efficient proximal operator. As a result, directly comparing the two objectives under our original framework would be unfair.

To ensure a fair comparison, we follow the methodology of Chhaya et al. (2020), which utilizes the Global Optimization Toolbox from MATLAB. Specifically, we use the `patternsearch` solver to address both the standard and modified LASSO problems. In our experiments, the solver parameters are set as follows: `MaxFunctionEvaluations` = 1,000,000, and `MaxIterations` = 25,000. To quantify the approximation quality of the coreset algorithm, we utilize the relative error, defined as $\frac{|V_1 - V_2|}{|V_1|}$, where $V_1$ denotes the objective value on the full dataset and $V_2$ denotes the objective value on the coreset solution evaluated on the full dataset. The experiments are conducted on a machine equipped with an Intel(R) Core(TM) i5-14400F 16-core processor with 32GB of memory, and the implementation is executed in MATLAB R2018a.

We first use Algorithm 1 to construct the coreset, and then apply the `patternsearch` solver to solve both objective functions on the coreset samples. The experiments are conducted on mediamill dataset using the same coreset sizes and regularization parameters $\lambda$ as in Section 4. Each experiment is repeated 10 times, and we report the average results. As shown in Table 9, the sensitivity sampling algorithm for standard LASSO generally achieves a lower relative error compared to the modified objective at the same sparsity level.

# D USE OF LARGE LANGUAGE MODELS (LLMS)

No large language models were used in the ideation or writing of this paper.

Table 2: Comparison results of loss, runtime, and sparsity on CTs dataset ($n = 53{,}500$, $d = 386$) for varying coreset sizes at $\lambda = \{1, 5, 10\}$.

| Lambda | Metrics | Algorithms | Coreset Sizes | | | | | |
|---|---|---|---|---|---|---|---|---|
| | | | $1d$ | $2d$ | $5d$ | $10d$ | $15d$ | $20d$ |
| $\lambda = 1$ | Loss | LASSO | | | 50.52±0.42 | | | |
| | | **LASSO-Sens** | 53.62±5.76 | 50.89±0.66 | 50.59±0.03 | 50.57±0.03 | 50.57±0.02 | 50.58±0.02 |
| | | LASSO-Uniform | 84.57±22.59 | 56.56±7.58 | 51.36±0.45 | 50.87±0.20 | 50.78±0.16 | 50.71±0.14 |
| | Time (s) | LASSO | | | 507.01 | | | |
| | | **LASSO-Sens** | 9.32 | 13.17 | 21.20 | 36.72 | 68.46 | 121.51 |
| | | LASSO-Uniform | 8.93 | 13.25 | 21.75 | 38.00 | 72.04 | 132.07 |
| | Sparsity | LASSO | | | 229 | | | |
| | | **LASSO-Sens** | 339 | 255 | 236 | 234 | 223 | 226 |
| | | LASSO-Uniform | 341 | 265 | 228 | 224 | 220 | 230 |
| $\lambda = 5$ | Loss | LASSO | | | 252.45±0.10 | | | |
| | | **LASSO-Sens** | 270.90±14.83 | 254.70±0.96 | 253.04±0.30 | 252.71±0.12 | 252.63±0.07 | 252.62±0.08 |
| | | LASSO-Uniform | 323.35±40.59 | 263.83±6.21 | 254.30±0.90 | 253.30±0.51 | 253.08±0.47 | 252.89±0.30 |
| | Time (s) | LASSO | | | 502.08 | | | |
| | | **LASSO-Sens** | 9.51 | 13.56 | 20.96 | 35.76 | 66.54 | 118.12 |
| | | LASSO-Uniform | 8.95 | 13.40 | 21.24 | 37.12 | 74.12 | 129.55 |
| | Sparsity | LASSO | | | 156 | | | |
| | | **LASSO-Sens** | 176 | 169 | 161 | 160 | 154 | 159 |
| | | LASSO-Uniform | 183 | 172 | 165 | 153 | 173 | 162 |
| $\lambda = 10$ | Loss | LASSO | | | 504.83±0.19 | | | |
| | | **LASSO-Sens** | 548.67±20.11 | 512.16±3.30 | 506.54±0.91 | 505.73±0.50 | 505.60±0.38 | 505.30±0.24 |
| | | LASSO-Uniform | 656.63±79.21 | 534.78±22.85 | 510.78±3.99 | 506.99±1.49 | 506.10±1.00 | 505.75±0.65 |
| | Time (s) | LASSO | | | 517.13 | | | |
| | | **LASSO-Sens** | 9.26 | 13.34 | 20.65 | 36.70 | 70.58 | 110.78 |
| | | LASSO-Uniform | 8.92 | 13.42 | 21.80 | 37.16 | 73.78 | 124.45 |
| | Sparsity | LASSO | | | 156 | | | |
| | | **LASSO-Sens** | 176 | 163 | 156 | 152 | 154 | 158 |
| | | LASSO-Uniform | 183 | 170 | 155 | 154 | 157 | 154 |

Table 3: Comparison results of loss, runtime, and sparsity on Synthetic dataset ($n = 10,000, d = 200$) for varying coreset sizes at $\lambda = \{0.5, 1, 5, 10\}$.

| Lambda | Metrics | Algorithm | Coreset Sizes | | | | | |
|---|---|---|---|---|---|---|---|---|
| | | | $1d$ | $2d$ | $5d$ | $10d$ | $15d$ | $20d$ |
| $\lambda = 0.5$ | Loss | LASSO | | | 11.80±0.12 | | | |
| | | **LASSO-Sens** | 12.78±0.49 | 12.89±0.48 | 12.55±0.39 | 12.28±0.24 | 12.23±0.32 | 12.25±0.23 |
| | | LASSO-Uniform | 14.75±0.14 | 14.76±0.15 | 14.55±0.33 | 14.30±0.47 | 13.87±0.50 | 13.36±0.49 |
| | Time (s) | LASSO | | | 72.76 | | | |
| | | **LASSO-Sens** | 5.74 | 7.40 | 9.27 | 14.49 | 18.24 | 21.06 |
| | | LASSO-Uniform | 5.86 | 7.99 | 10.17 | 15.02 | 19.81 | 23.69 |
| | Sparsity | LASSO | | | 40 | | | |
| | | **LASSO-Sens** | 38 | 33 | 35 | 35 | 36 | 38 |
| | | LASSO-Uniform | 28 | 28 | 28 | 28 | 30 | 32 |
| $\lambda = 1$ | Loss | LASSO | | | 20.85±0.15 | | | |
| | | **LASSO-Sens** | 20.66±0.15 | 20.51±0.17 | 20.51±0.19 | 20.57±0.14 | 20.67±0.13 | 20.69±0.12 |
| | | LASSO-Uniform | 20.90±0.11 | 20.96±0.12 | 21.05±0.16 | 20.92±0.15 | 20.79±0.12 | 20.60±0.22 |
| | Time (s) | LASSO | | | 78.34 | | | |
| | | **LASSO-Sens** | 5.68 | 7.43 | 9.28 | 14.31 | 18.65 | 21.29 |
| | | LASSO-Uniform | 5.93 | 7.99 | 9.76 | 14.84 | 20.46 | 23.11 |
| | Sparsity | LASSO | | | 28 | | | |
| | | **LASSO-Sens** | 31 | 28 | 29 | 28 | 28 | 28 |
| | | LASSO-Uniform | 27 | 27 | 28 | 29 | 28 | 31 |
| $\lambda = 5$ | Loss | LASSO | | | 69.15±0.20 | | | |
| | | **LASSO-Sens** | 69.31±0.14 | 69.18±0.03 | 69.16±0.01 | 69.15±0.00 | 69.15±0.00 | 69.15±0.00 |
| | | LASSO-Uniform | 69.95±1.68 | 70.62±1.84 | 70.65±1.68 | 69.35±0.49 | 69.28±0.33 | 69.22±0.24 |
| | Time (s) | LASSO | | | 72.55 | | | |
| | | **LASSO-Sens** | 5.64 | 7.41 | 9.35 | 14.48 | 18.57 | 21.49 |
| | | LASSO-Uniform | 5.90 | 7.82 | 9.81 | 15.86 | 20.39 | 24.88 |
| | Sparsity | LASSO | | | 27 | | | |
| | | **LASSO-Sens** | 28 | 27 | 27 | 27 | 27 | 27 |
| | | LASSO-Uniform | 27 | 27 | 27 | 27 | 27 | 27 |
| $\lambda = 10$ | Loss | LASSO | | | 129.53±0.08 | | | |
| | | **LASSO-Sens** | 130.27±0.58 | 129.66±0.07 | 129.56±0.03 | 129.54±0.02 | 129.53±0.01 | 129.53±0.01 |
| | | LASSO-Uniform | 131.43±3.10 | 130.78±1.83 | 129.55±0.03 | 129.53±0.01 | 129.53±0.01 | 129.53±0.01 |
| | Time (s) | LASSO | | | 77.02 | | | |
| | | **LASSO-Sens** | 5.67 | 7.31 | 9.27 | 14.60 | 18.09 | 20.92 |
| | | LASSO-Uniform | 5.85 | 7.92 | 9.96 | 15.07 | 19.35 | 24.56 |
| | Sparsity | LASSO | | | 27 | | | |
| | | **LASSO-Sens** | 29 | 27 | 27 | 27 | 27 | 27 |
| | | LASSO-Uniform | 27 | 27 | 27 | 27 | 27 | 27 |

Table 4: Comparison results of loss, runtime, and sparsity on mediamill dataset ($n = 30,993, d = 120$) for varying coreset sizes at $\lambda = \{0.5, 1, 5, 10\}$.

| Lambda | Metrics | Algorithms | Coreset Sizes | | | | | |
|---|---|---|---|---|---|---|---|---|
| | | | $1d$ | $2d$ | $5d$ | $10d$ | $15d$ | $20d$ |
| $\lambda = 0.5$ | Loss | LASSO | | | 7.35±0.15 | | | |
| | | **LASSO-Sens** | 7.68±0.14 | 7.41±0.03 | 7.37±0.01 | 7.36±0.00 | 7.36±0.00 | 7.35±0.00 |
| | | LASSO-Uniform | 7.89±0.42 | 7.47±0.04 | 7.38±0.01 | 7.37±0.01 | 7.36±0.01 | 7.36±0.00 |
| | Time (s) | LASSO | | | 148.64 | | | |
| | | **LASSO-Sens** | 4.01 | 5.05 | 7.70 | 11.09 | 2.95 | 4.03 |
| | | LASSO-Uniform | 3.90 | 4.99 | 7.56 | 10.81 | 2.92 | 3.97 |
| | Sparsity | LASSO | | | 59 | | | |
| | | **LASSO-Sens** | 51 | 53 | 51 | 56 | 57 | 55 |
| | | LASSO-Uniform | 46 | 50 | 53 | 55 | 57 | 54 |
| $\lambda = 1$ | Loss | LASSO | | | 14.68±0.18 | | | |
| | | **LASSO-Sens** | 15.50±0.35 | 14.87±0.07 | 14.74±0.03 | 14.71±0.01 | 14.70±0.01 | 14.69±0.01 |
| | | LASSO-Uniform | 16.23±0.71 | 15.12±0.25 | 14.79±0.04 | 14.73±0.03 | 14.70±0.01 | 14.69±0.00 |
| | Time (s) | LASSO | | | 149.90 | | | |
| | | **LASSO-Sens** | 3.90 | 5.33 | 7.70 | 11.06 | 3.01 | 4.06 |
| | | LASSO-Uniform | 3.70 | 5.12 | 7.55 | 11.01 | 2.95 | 4.14 |
| | Sparsity | LASSO | | | 58 | | | |
| | | **LASSO-Sens** | 50 | 52 | 54 | 55 | 54 | 58 |
| | | LASSO-Uniform | 48 | 52 | 52 | 51 | 53 | 57 |
| $\lambda = 5$ | Loss | LASSO | | | 72.86±0.24 | | | |
| | | **LASSO-Sens** | 78.18±2.81 | 74.88±1.03 | 73.43±0.30 | 73.16±0.18 | 73.05±0.19 | 72.92±0.01 |
| | | LASSO-Uniform | 81.81±3.94 | 76.47±1.76 | 73.66±0.61 | 73.37±0.49 | 73.15±0.28 | 73.07±0.29 |
| | Time (s) | LASSO | | | 151.54 | | | |
| | | **LASSO-Sens** | 3.78 | 4.77 | 8.00 | 11.08 | 3.04 | 4.03 |
| | | LASSO-Uniform | 3.63 | 4.82 | 7.56 | 10.98 | 2.97 | 4.03 |
| | Sparsity | LASSO | | | 58 | | | |
| | | **LASSO-Sens** | 40 | 45 | 48 | 49 | 55 | 54 |
| | | LASSO-Uniform | 36 | 40 | 47 | 49 | 52 | 54 |
| $\lambda = 10$ | Loss | LASSO | | | 143.91±0.04 | | | |
| | | **LASSO-Sens** | 152.40±3.84 | 147.86±2.32 | 145.13±0.82 | 144.42±0.34 | 144.18±0.22 | 144.47±0.45 |
| | | LASSO-Uniform | 160.69±10.16 | 150.62±4.67 | 146.29±1.73 | 144.63±0.61 | 144.97±1.12 | 144.73±0.27 |
| | Time (s) | LASSO | | | 155.54 | | | |
| | | **LASSO-Sens** | 3.75 | 4.94 | 8.27 | 10.86 | 3.03 | 4.26 |
| | | LASSO-Uniform | 3.58 | 4.81 | 8.35 | 10.75 | 2.98 | 4.22 |
| | Sparsity | LASSO | | | 47 | | | |
| | | **LASSO-Sens** | 32 | 39 | 40 | 43 | 44 | 44 |
| | | LASSO-Uniform | 32 | 36 | 37 | 42 | 43 | 43 |

Table 5: Comparison results of loss, runtime, and sparsity on mnist8m datasets ($n = 8,100,000, d = 784$) for varying coreset sizes at $\lambda = \{0.5, 1, 5, 10\}$. If an algorithm fails to output a solution within 48 hours, the metrics are marked as $48 > h$.

| Lambda | Metrics | Algorithms | 1d | 2d | 5d | 10d | 15d | 20d |
|---|---|---|---|---|---|---|---|---|
| $\lambda = 0.5$ | Loss | LASSO | | | $48 > h$ | | | |
| | | **LASSO-Sens** | 1.27E7 ± 1.02E7 | 3.30E4 ± 7.71E3 | 1.64E4 ± 4.03E2 | 1.45E4 ± 3.06E2 | 1.43E4 ± 1.80E2 | 1.37E4 ± 1.16E2 |
| | | LASSO-Uniform | 5.83$E$8 ± 2.73$E$8 | 1.79E8 ± 4.25E7 | 3.59E7 ± 3.63E7 | 6.53E6 ± 2.90E6 | 4.46E6 ± 2.85E6 | 3.00E6 ± 1.09E6 |
| | Time (s) | LASSO | | | $48 > h$ | | | |
| | | **LASSO-Sens** | 304.32 | 314.23 | 370.11 | 512.73 | 703.91 | 859.22 |
| | | LASSO-Uniform | 19.39 | 30.61 | 58.05 | 184.22 | 336.58 | 459.11 |
| | Sparsity | LASSO | | | $48 > h$ | | | |
| | | **LASSO-Sens** | 780 | 776 | 770 | 770 | 763 | 760 |
| | | LASSO-Uniform | 704 | 712 | 725 | 728 | 735 | 735 |
| $\lambda = 1$ | Loss | LASSO | | | $48 > h$ | | | |
| | | **LASSO-Sens** | 1.61E7 ± 1.04E7 | 3.46E4 ± 8.18E3 | 1.68E4 ± 5.27E2 | 1.48E4 ± 2.05E2 | 1.41E4 ± 1.38E2 | 1.38E4 ± 7.54E1 |
| | | LASSO-Uniform | 4.77E8 ± 5.68E7 | 1.35E8 ± 6.33E7 | 1.80E7 ± 3.20E6 | 9.35E6 ± 4.36E6 | 5.37E6 ± 2.21E6 | 3.04E6 ± 1.13E6 |
| | Time (s) | LASSO | | | $48 > h$ | | | |
| | | **LASSO-Sens** | 302.37 | 316.26 | 366.99 | 522.60 | 724.51 | 838.05 |
| | | LASSO-Uniform | 20.25 | 30.3 | 58.01 | 187.43 | 352.15 | 463.57 |
| | Sparsity | LASSO | | | $48 > h$ | | | |
| | | **LASSO-Sens** | 778 | 774 | 768 | 763 | 765 | 758 |
| | | LASSO-Uniform | 705 | 709 | 731 | 737 | 737 | 729 |
| $\lambda = 5$ | Loss | LASSO | | | $48 > h$ | | | |
| | | **LASSO-Sens** | 1.39E7 ± 4.50E6 | 2.87E4 ± 4.21E3 | 1.72E4 ± 4.57E2 | 1.49E4 ± 2.52E2 | 1.45E4 ± 1.88E2 | 1.42E4 ± 1.49E2 |
| | | LASSO-Uniform | 5.98E8 ± 7.60E7 | 1.23E8 ± 3.26E7 | 2.15E7 ± 7.61E6 | 9.22E6 ± 2.22E6 | 3.99E6 ± 1.09E6 | 3.05E6 ± 9.08E5 |
| | Time (s) | LASSO | | | $48 > h$ | | | |
| | | **LASSO-Sens** | 299.24 | 317.45 | 368.02 | 527.51 | 708.16 | 875.08 |
| | | LASSO-Uniform | 19.32 | 26.66 | 57.07 | 186.32 | 341.58 | 401.96 |
| | Sparsity | LASSO | | | $48 > h$ | | | |
| | | **LASSO-Sens** | 780 | 776 | 770 | 767 | 765 | 763 |
| | | LASSO-Uniform | 691 | 714 | 729 | 735 | 730 | 741 |
| $\lambda = 10$ | Loss | LASSO | | | $48 > h$ | | | |
| | | **LASSO-Sens** | 1.10E7 ± 8.22E6 | 9.47E4 ± 1.27E5 | 1.73E4 ± 1.06E2 | 1.55E4 ± 1.54E2 | 1.49E4 ± 3.28E1 | 1.47E4 ± 8.80E1 |
| | | LASSO-Uniform | 4.96E8 ± 1.81E8 | 1.25E8 ± 5.99E7 | 1.58E7 ± 5.71E6 | 8.34E6 ± 1.86E6 | 3.99E6 ± 9.91E5 | 3.50E6 ± 6.16E5 |
| | Time (s) | LASSO | | | $48 > h$ | | | |
| | | **LASSO-Sens** | 302.45 | 318.28 | 363.87 | 528.66 | 720.44 | 870.80 |
| | | LASSO-Uniform | 19.95 | 24.2 | 59.64 | 187.25 | 327.57 | 513.41 |
| | Sparsity | LASSO | | | $48 > h$ | | | |
| | | **LASSO-Sens** | 781 | 777 | 769 | 769 | 765 | 763 |
| | | LASSO-Uniform | 700 | 713 | 730 | 736 | 728 | 737 |

Table 6: Comparison of the sketching algorithm and LASSO-Sens on the CTs dataset ($n = 53,500, d = 386$).

| Lambda | Metrics | Algorithms | 1d | 2d | 5d | 10d | 15d | 20d |
|---|---|---|---|---|---|---|---|---|
| $\lambda = 0.5$ | Loss | **LASSO-Sens** | 24.11±0.30 | 23.99±0.01 | 23.98±0.00 | 23.97±0.00 | 23.97±0.00 | 23.97±0.00 |
| | | Sketching | 24.13±0.03 | 23.99±0.00 | 23.98±0.00 | 23.97±0.00 | 23.97±0.00 | 23.97±0.00 |
| | Time (s) | **LASSO-Sens** | 7.21 | 7.18 | 9.66 | 13.58 | 17.68 | 22.62 |
| | | Sketching | 5.91 | 6.61 | 10.49 | 16.05 | 21.37 | 27.46 |
| | Sparsity | **LASSO-Sens** | 383 | 359 | 322 | 317 | 314 | 316 |
| | | Sketching | 382 | 360 | 327 | 319 | 313 | 309 |
| $\lambda = 1$ | Loss | **LASSO-Sens** | 48.10±0.75 | 47.97±0.02 | 47.93±0.00 | 47.93±0.00 | 47.93±0.00 | 47.92±0.00 |
| | | Sketching | 48.17±0.06 | 47.98±0.01 | 47.94±0.00 | 47.93±0.04 | 47.93±0.02 | 47.93±0.03 |
| | Time (s) | **LASSO-Sens** | 6.83 | 7.26 | 9.95 | 13.38 | 18.51 | 22.27 |
| | | Sketching | 6.11 | 6.74 | 10.40 | 16.12 | 21.54 | 27.23 |
| | Sparsity | **LASSO-Sens** | 345 | 241 | 219 | 215 | 217 | 215 |
| | | Sketching | 327 | 243 | 222 | 214 | 214 | 216 |
| $\lambda = 5$ | Loss | **LASSO-Sens** | 242.48±1.45 | 240.20±0.62 | 239.77±0.11 | 239.65±0.06 | 239.61±0.05 | 239.61±0.04 |
| | | Sketching | 243.89±1.67 | 240.39±0.14 | 239.86±0.05 | 239.68±0.05 | 239.65±0.05 | 239.64±0.02 |
| | Time (s) | **LASSO-Sens** | 7.13 | 7.28 | 9.90 | 13.85 | 18.00 | 21.96 |
| | | Sketching | 6.12 | 6.79 | 10.20 | 16.36 | 21.52 | 26.74 |
| | Sparsity | **LASSO-Sens** | 179 | 168 | 156 | 154 | 158 | 157 |
| | | Sketching | 174 | 166 | 159 | 155 | 159 | 158 |
| $\lambda = 10$ | Loss | **LASSO-Sens** | 495.54±2.59 | 481.58±1.01 | 479.82±0.69 | 479.18±0.22 | 479.17±0.14 | 479.00±0.23 |
| | | Sketching | 495.48±2.57 | 482.36±1.07 | 479.98±0.24 | 479.52±0.14 | 479.33±0.17 | 479.28±0.05 |
| | Time (s) | **LASSO-Sens** | 7.21 | 7.29 | 9.77 | 13.34 | 18.26 | 21.74 |
| | | Sketching | 6.09 | 7.05 | 10.41 | 16.14 | 21.51 | 26.85 |
| | Sparsity | **LASSO-Sens** | 169 | 153 | 157 | 153 | 155 | 153 |
| | | Sketching | 170 | 159 | 149 | 149 | 156 | 153 |

Table 7: Comparison of the sketching algorithm and LASSO-Sens on the Synthetic dataset ($n = 10,000, d = 200$).

| Lambda | Metrics | Algorithm | Coreset Sizes | | | | | |
|---|---|---|---|---|---|---|---|---|
| | | | $1d$ | $2d$ | $5d$ | $10d$ | $15d$ | $20d$ |
| $\lambda = 0.5$ | Loss | **LASSO-Sens** | 14.24±0.41 | 14.02±0.30 | 13.27±0.59 | 12.83±0.43 | 12.93±0.26 | 13.12±0.10 |
| | | Sketching | 25.82±4.57 | 17.15±0.81 | 14.12±0.51 | 13.52±0.20 | 13.39±0.12 | 13.19±0.16 |
| | Time (s) | **LASSO-Sens** | 4.23 | 4.77 | 5.55 | 6.19 | 7.60 | 8.21 |
| | | Sketching | 3.84 | 4.96 | 5.72 | 6.38 | 7.93 | 9.21 |
| | Sparsity | **LASSO-Sens** | 40 | 34 | 38 | 39 | 40 | 40 |
| | | Sketching | 101 | 106 | 113 | 111 | 111 | 108 |
| $\lambda = 1$ | Loss | **LASSO-Sens** | 21.41±0.17 | 21.42±0.22 | 21.53±0.06 | 21.47±0.14 | 21.57±0.12 | 21.61±0.06 |
| | | Sketching | 31.52±1.92 | 24.89±0.22 | 22.90±0.18 | 22.31±0.08 | 22.23±0.03 | 22.14±0.05 |
| | Time (s) | **LASSO-Sens** | 4.03 | 4.98 | 4.88 | 6.34 | 7.67 | 7.99 |
| | | Sketching | 4.02 | 4.99 | 5.14 | 6.51 | 7.85 | 9.11 |
| | Sparsity | **LASSO-Sens** | 35 | 29 | 28 | 27 | 26 | 26 |
| | | Sketching | 96 | 95 | 94 | 92 | 90 | 90 |
| $\lambda = 5$ | Loss | **LASSO-Sens** | 66.78±0.08 | 66.77±0.01 | 66.76±0.00 | 66.76±0.00 | 66.76±0.00 | 66.76±0.00 |
| | | Sketching | 72.24±1.33 | 69.02±0.43 | 67.58±0.19 | 67.09±0.05 | 67.00±0.06 | 66.88±0.02 |
| | Time (s) | **LASSO-Sens** | 3.90 | 5.40 | 4.81 | 6.39 | 7.64 | 8.04 |
| | | Sketching | 3.82 | 5.29 | 5.18 | 6.64 | 7.84 | 9.20 |
| | Sparsity | **LASSO-Sens** | 29 | 25 | 24 | 24 | 24 | 24 |
| | | Sketching | 80 | 78 | 70 | 58 | 55 | 49 |
| $\lambda = 10$ | Loss | **LASSO-Sens** | 122.99±0.28 | 122.87±0.09 | 122.81±0.02 | 122.82±0.01 | 122.82±0.01 | 122.83±0.01 |
| | | Sketching | 127.49±0.81 | 124.60±0.35 | 123.48±0.16 | 123.02±0.04 | 122.96±0.02 | 122.92±0.03 |
| | Time (s) | **LASSO-Sens** | 4.13 | 5.30 | 4.83 | 6.48 | 7.63 | 8.07 |
| | | Sketching | 3.95 | 5.20 | 5.11 | 6.65 | 7.71 | 9.16 |
| | Sparsity | **LASSO-Sens** | 30 | 25 | 24 | 24 | 24 | 24 |
| | | Sketching | 65 | 59 | 47 | 37 | 33 | 27 |

Table 8: Comparison of the sketching algorithm and LASSO-Sens on the mediamill dataset ($n = 30,993, d = 120$).

| Lambda | Metrics | Algorithms | Coreset Sizes | | | | | |
|---|---|---|---|---|---|---|---|---|
| | | | $1d$ | $2d$ | $5d$ | $10d$ | $15d$ | $20d$ |
| $\lambda = 0.5$ | Loss | **LASSO-Sens** | 8.40±0.43 | 8.19±0.04 | 8.17±0.02 | 8.15±0.01 | 8.15±0.00 | 8.15±0.00 |
| | | Sketching | 8.54±0.15 | 8.25±0.04 | 8.18±0.01 | 8.16±0.00 | 8.15±0.00 | 8.15±0.00 |
| | Time (s) | **LASSO-Sens** | 2.57 | 3.37 | 4.69 | 4.94 | 4.69 | 5.81 |
| | | Sketching | 2.14 | 3.28 | 4.59 | 5.07 | 5.13 | 5.82 |
| | Sparsity | **LASSO-Sens** | 58 | 60 | 62 | 63 | 61 | 61 |
| | | Sketching | 57 | 60 | 60 | 60 | 62 | 63 |
| $\lambda = 1$ | Loss | **LASSO-Sens** | 16.77±0.23 | 16.34±0.16 | 16.27±0.03 | 16.24±0.02 | 16.23±0.02 | 16.22±0.01 |
| | | Sketching | 17.16±0.45 | 16.46±0.07 | 16.29±0.04 | 16.25±0.01 | 16.25±0.01 | 16.24±0.01 |
| | Time (s) | **LASSO-Sens** | 2.47 | 3.68 | 4.68 | 4.91 | 4.48 | 5.82 |
| | | Sketching | 2.13 | 3.07 | 4.61 | 4.91 | 5.30 | 5.78 |
| | Sparsity | **LASSO-Sens** | 57 | 58 | 59 | 61 | 57 | 60 |
| | | Sketching | 54 | 59 | 62 | 59 | 61 | 60 |
| $\lambda = 5$ | Loss | **LASSO-Sens** | 83.62±4.37 | 81.00±0.93 | 80.30±0.18 | 79.95±0.18 | 80.07±0.08 | 80.01±0.09 |
| | | Sketching | 87.07±1.31 | 82.66±0.93 | 80.57±0.37 | 80.24±0.11 | 80.26±0.17 | 80.20±0.11 |
| | Time (s) | **LASSO-Sens** | 2.46 | 3.45 | 4.71 | 4.93 | 4.49 | 5.82 |
| | | Sketching | 2.26 | 3.30 | 4.64 | 5.02 | 5.20 | 5.90 |
| | Sparsity | **LASSO-Sens** | 44 | 51 | 51 | 51 | 51 | 51 |
| | | Sketching | 46 | 49 | 49 | 53 | 51 | 52 |
| $\lambda = 10$ | Loss | **LASSO-Sens** | 163.26±8.88 | 160.69±5.25 | 159.24±0.35 | 158.89±0.50 | 158.91±0.34 | 158.65±0.40 |
| | | Sketching | 177.47±5.03 | 163.25±0.83 | 160.31±0.97 | 159.24±0.37 | 159.08±0.26 | 159.03±0.14 |
| | Time (s) | **LASSO-Sens** | 2.56 | 3.45 | 4.65 | 4.90 | 4.47 | 5.81 |
| | | Sketching | 2.11 | 3.23 | 4.50 | 4.80 | 5.31 | 5.83 |
| | Sparsity | **LASSO-Sens** | 40 | 44 | 48 | 51 | 49 | 49 |
| | | Sketching | 41 | 46 | 51 | 49 | 49 | 49 |

Table 9: Comparison of sensitivity sampling applied to standard and modified LASSO objectives on mediamill dataset ($n = 30,993$, $d = 120$).

| Lambda | Metrics | Algorithms | Coreset Sizes | | | | | |
|---|---|---|---|---|---|---|---|---|
| | | | $1d$ | $2d$ | $5d$ | $10d$ | $15d$ | $20d$ |
| $\lambda = 1$ | Relative error | **LASSO** | 0.1803±0.1512 | **0.1059±0.0616** | **0.0417±0.0291** | **0.0430±0.0372** | **0.0197±0.0057** | **0.0243±0.0215** |
| | | modified LASSO | **0.1016±0.0668** | 0.1143±0.0852 | 0.0972±0.0944 | 0.0800±0.0626 | 0.0434±0.0405 | 0.0543±0.0491 |
| | Time (s) | **LASSO** | 8.04 | 13.49 | 15.09 | 16.90 | 18.68 | 22.23 |
| | | modified LASSO | 7.81 | 13.25 | 14.84 | 16.73 | 18.49 | 21.97 |
| | Sparsity | **LASSO** | | | | 120 | | |
| | | modified LASSO | | | | 120 | | |
| $\lambda = 5$ | Relative error | **LASSO** | **0.1267±0.1121** | **0.0646±0.0529** | **0.0520±0.0321** | **0.0306±0.0334** | **0.0300±0.0423** | **0.0170±0.0145** |
| | | modified LASSO | 0.1402±0.2204 | 0.1340±0.1017 | 0.1166±0.1329 | 0.0809±0.0547 | 0.0485±0.0522 | 0.0689±0.0916 |
| | Time (s) | **LASSO** | 8.10 | 13.54 | 15.17 | 16.74 | 18.93 | 22.19 |
| | | modified LASSO | 7.94 | 13.34 | 15.04 | 16.68 | 18.83 | 21.97 |
| | Sparsity | **LASSO** | | | | 120 | | |
| | | modified LASSO | | | | 120 | | |
| $\lambda = 10$ | Relative error | **LASSO** | 0.1387±0.0985 | 0.0828±0.0443 | 0.0448±0.0385 | **0.0345±0.0239** | **0.0314±0.0283** | **0.0221±0.0154** |
| | | modified LASSO | **0.1046±0.1057** | **0.0730±0.0449** | **0.0296±0.0333** | 0.0905±0.0462 | 0.0331±0.0328 | 0.0399±0.0313 |
| | Time (s) | **LASSO** | 8.07 | 13.54 | 15.12 | 16.68 | 18.75 | 22.10 |
| | | modified LASSO | 7.89 | 13.37 | 14.96 | 16.67 | 18.64 | 21.98 |
| | Sparsity | **LASSO** | | | | 120 | | |
| | | modified LASSO | | | | 120 | | |

