# OpenReview forum: "On Coreset for LASSO Regression Problem with Sensitivity Sampling"
_ICLR.cc/2026/Conference — ICLR 2026 Poster_

### Official Review · Reviewer_VhuG · 2025-10-30

**Soundness:** 3
**Presentation:** 4
**Contribution:** 4
**Rating:** 8
**Confidence:** 3

**Summary:**

This paper introduces ``LASSO-Sens``, the first coreset construction method for the standard LASSO regression problem based on sensitivity sampling.
The key challenge in applying coreset techniques to LASSO stems from the complex, non-smooth geometry of the function space induced by the l1 penalty, which complicates standard analysis . The authors overcome this barrier by developing a localized empirical process method that effectively decomposes the function space into independent residual and l1 penalty components.
This new analysis yields a provably tight coreset size up to logarithmic terms. The authors confirm this bound is nearly optimal by providing a matching lower bound .

Experimental results supplement the theory, demonstrating that ``LASSO-Sens`` is significantly more efficient (four times faster) than the standard LASSO solver and substantially outperforms a uniform sampling coreset baseline, all while maintaining high solution quality and sparsity.

**Strengths:**

**High-Quality Presentation:** The paper is exceptionally well-written and easy to follow. The authors do an excellent job of situating their work within the existing literature, clearly motivating their approach and making the theoretical breakthroughs highly compelling.

**Significant Theoretical Contribution:** The primary contribution, a comprehensive and provably near-tight coreset size bound, is a significant theoretical achievement. Overcoming the noted analytical hurdles of bounding the sampling error for the LASSO objective is a noteworthy accomplishment. *Though, I am unfamiliar with this field and defer to other reviewers to confirm the novelty of this result within the broader literature.*

**Comprehensive Analysis:** The paper's theoretical claims are well-supported, complete with a matching lower bound in the relevant regime, which confirms the near-optimality of the proposed method. *Again, I defer to other reviewers to confirm that the relationship between $\lambda$ and $d$ is common for comparing the here proven upper and lower bounds.*

**Weaknesses:**

The primary weakness of the paper lies in its experimental validation. While the theory is the main focus, the empirical results section feels underdeveloped and, in some cases, seems to contradict the narrative that LASSO-Sens is the superior practical approach.

**Overstated Claims of Empirical Superiority:** The text emphasizes multiplicative speedups of ``LASSO-Sens`` over the full LASSO procedure. However, this comparison obscures a more critical one: the performance against the LASSO-Uniform baseline.

**LASSO-Sens vs. LASSO-Uniform:** Across many of the provided plots (Figures 1, 2, and appendix figures), ``LASSO-Sens`` does not demonstrate a clear, significant advantage over ``LASSO-Uniform`` in terms of final loss . In several instances (e.g., Synthetic $\lambda = 0.5, 1$), ``LASSO-Uniform`` even appears to achieve a better loss.

**Lack of Statistical Rigor:** The plots do not include statistical error bars, and the text does not clarify if the results are from a single replication or averaged over many trials. Without this, it is impossible to determine if the small observed differences between ``LASSO-Sens`` and ``LASSO-Uniform`` are statistically significant or simply noise. Given that ``LASSO-Uniform`` is a lighter-weight approach (avoiding the sensitivity score computation), its comparable effectiveness in these experiments weakens the practical argument for the proposed method.

**Questions:**

**Statistical Significance:** Are the results in Figures 1 and 2 (and appendix figures) from a single run or averaged over the 10 trials mentioned? Could you please add error bars (e.g., standard error or 95% CIs) to the plots to allow for a proper statistical comparison between ``LASSO-Sens`` and ``LASSO-Uniform``?

**Practical Justification:** Given that ``LASSO-Uniform`` performs comparably, and sometimes better, than ``LASSO-Sens`` in the provided experiments, could you further emphasize the empirical justification for using the more complex sensitivity sampling method? The theoretical advantage is clear, but the practical advantage over this strong, simple baseline is not.

**Cost of Sensitivities:** Do the runtimes reported for ``LASSO-Sens`` (e.g., in Table 1) include the pre-processing time required to compute the sensitivity scores? A clear breakdown of the (sampling vs. solving) costs for both coreset methods would be essential for a fair comparison.

---

> ### Author Response · Authors · 2025-11-21
> **Response to Reviewer VhuG**
>
> **Question 1**. Statistical Significance: Are the results in Figures 1 and 2 (and appendix figures) from a single run or averaged over the 10 trials mentioned? Could you please add error bars (e.g., standard error or 95% CIs) to the plots to allow for a proper statistical comparison between LASSO-Sens and LASSO-Uniform?
>
> Response. Thank you for pointing this out. All results presented in Figures 1 and 2 are averaged over 10 independent trials, as described in the experimental setup. We apologize for the omission of error bars in the original submission. In the revised version, we have included all relevant plots of Figures 1 and 2 to include standard error bars across trials. As shown in Figures 1 and 2, the LASSO-Sens algorithm maintains high-quality in metric of loss across almost all parameter settings.
>
> **Question 2**. Practical Justification: Given that LASSO-Uniform performs comparably, and sometimes better, than LASSO-Sens in the provided experiments, could you further emphasize the empirical justification for using the more complex sensitivity sampling method? The theoretical advantage is clear, but the practical advantage over this strong, simple baseline is not.
>
> Response. Thank you for the problem. We note that in the synthetic dataset under strong regularization ($\lambda=1$) and large coreset size ($20d$), LASSO-Uniform slightly outperforms LASSO-Sens in loss. As noted by Chhaya et al., (2020), the Synthetic dataset has an extremely skewed leverage score distribution: a small portion of the samples have very high leverage scores, while the rest have extremely low leverage scores.
>
> In such settings, LASSO-Sens tends to prioritize high-leverage samples, which is beneficial when the coreset size is small. Indeed, across most small to moderate coreset sizes (e.g., $1d$ to $10d$), LASSO-Sens generally outperforms LASSO-Uniform. However, when the coreset size becomes large (e.g., $20d$), the over-representation of high-leverage points may lead to under-sampling of the low-leverage region, reducing the diversity of the selected subset and negatively impacting the overall loss. In contrast, uniform sampling may select a more balanced subset under such conditions, leading to slightly better empirical performance in this specific regime.
>
> On real-world datasets such as CTs, mediamill, and mnist8m, LASSO-Sens achieves lower regression loss than LASSO-Uniform across a wide range of $\lambda$ and coreset sizes. Furthermore, the two methods yield comparable sparsity levels when the coreset size exceeds $10d$. Overall, LASSO-Sens provides more stable performance across varying regimes and datasets.
>
>
> **Question 3**. Cost of Sensitivities: Do the runtimes reported for LASSO-Sens (e.g., in Table 1) include the pre-processing time required to compute the sensitivity scores? A clear breakdown of the (sampling vs. solving) costs for both coreset methods would be essential for a fair comparison.
>
> Response. Thanks for pointing this out. The runtimes reported for LASSO-Sens in Table 1 do include both the time required to compute the sensitivity scores (i.e., coreset construction) and the time to solve the LASSO problem on the coreset. Specifically, the running time of coreset construction is $O(\text{nnz}(A)\log n+d^3\log d\log(n/d))$, where $\text{nnz}(A)$ denotes the number of non-zero entries in the input matrix $A$. The running time of LASSO using FISTA method is $O(md/\sqrt{\epsilon})$, where $m=\tilde{O}(\epsilon^{-2}d\cdot((\log d)^3\cdot \min(1,\log d/\lambda^2)+ \log(1/ \delta)))$ is the coreset size. For the LASSO-Uniform, the coreset construction (i.e., uniform sampling) takes $O(n)$, and solving LASSO using FISTA also requires $O(md/\sqrt{\epsilon})$ time. We will report the execution times of the sampling and solving steps separately in the final version.

---

> > ### Comment · Reviewer_VhuG · 2025-11-25
> >
> > Thank you for addressing my questions and concerns. I maintain my accept recommendation.

---

### Official Review · Reviewer_FxuS · 2025-11-01

**Soundness:** 2
**Presentation:** 2
**Contribution:** 3
**Rating:** 4
**Confidence:** 3

**Summary:**

This paper studies coresets for standard LASSO (least-squares + $\ell_1$ penalty) using sensitivity sampling. The authors propose a sensitivity-based sampling algorithm for augmenting matrix A′ = [A −b] and localizing the function class into a residual $\ell_2$ part and an $\ell_1$‑penalty part, then, they apply empirical process and chaining tools to bound Gaussian diameter and metric entropy on the localized sets, and finally uses those bounds to show a coreset of size is smaller than linear regression.

**Strengths:**

- Localizing the analysis into residual $\ell_2$ and $\ell_1$ penalty components and applying Gaussian/chaining bounds on each piece is a fresh and appropriate way to tackle the complexity introduced by the $\ell_1$ term.

**Weaknesses:**

- A thorough discussion and comparison with the existing literature, such as Avron et al and Chhaya et al, is missing.

**Questions:**

1. In Theorem 7, why $\log(\frac{1}{\delta})$ is an additive term in the coreset size?

2. The lasso negative result in Chhaya et al seems to be general, that is, a smaller strong coreset or sketch or summarization is impossible for any $\lambda$, i.e., the claim is independent of how the coreset is constructed and then analyzed. Can you clarify how your result negates this claim and if it does not negate, then how do you give a strong coreset guarantee on lasso.

3. How is the coreset size using the localization of the function class related to other standard regularized regressions, such as ridge (Avron et al) and modified lasso?

---

> ### Author Response · Authors · 2025-11-21
> **Response to Reviewer FxuS**
>
> **Weakness 1**. A thorough discussion and comparison with the existing literature, such as Avron et al and Chhaya et al, is missing.
>
> Response: Thank you for pointing this out. In revised version, we update a detailed discussion of these works in the “Other Related Work” section.
>
> **Question 1**. In Theorem 7, why $log(1/\delta)$ is an additive term in the coreset size?
>
> Response. Thank you for the insightful question. We are sorry for the confusion. The additive term of $log(1/\delta)$ is a typo in proof Theorem 7 (see Appendix B). The rest of the proof remains correct with respect to the $log(1/\delta)$ term. In our analysis, the coreset size is $O(\alpha \cdot d)$, where the $\alpha$ is over-sampling parameter and it contains the term of $log(1/\delta)$. In revised version, the coreset size is $\tilde{O}\left(\frac{d(\log d)^3}{\epsilon^2}\cdot \min(1,\frac{\log d}{\lambda^2})+ \frac{d}{\epsilon^2} \log\frac{1}{\delta}\right)$.
>
> **Question 2**. The lasso negative result in Chhaya et al seems to be general, that is, a smaller strong coreset or sketch or summarization is impossible for any $\lambda$, i.e., the claim is independent of how the coreset is constructed and then analyzed. Can you clarify how your result negates this claim and if it does not negate, then how do you give a strong coreset guarantee on lasso.
>
> Response. Thank you for raising this important point. We apologize for the typo in our original coreset size expression that the placement of the $log(1/\delta)$ term was incorrect. The corrected coreset size bound is $\tilde{O}\left(\frac{d(\log d)^3}{\epsilon^2}\cdot \min(1, \frac{\log d}{\lambda^2}) + \frac{d}{\epsilon^2} \log\frac{1}{\delta}\right)$.
>
> This coreset is in fact consistent with the negative result presented in Corollary 4.1.1 of Chhaya et al. (2020). That work shows that for arbitrary $\lambda>0$. it is impossible to construct a strong coreset of size smaller than the optimal sized coreset for unregularized least squares regression. As $\lambda\rightarrow \infty$, our coreset size approaches the known optimal bound of $O(d/\epsilon^2\log(1/\delta))$, which does not contradict the result in Chhaya et al. (2020).
>
> **Question 3**. How is the coreset size using the localization of the function class related to other standard regularized regressions, such as ridge (Avron et al) and modified lasso?
>
> Response. Thank you for the thoughtful question. We believe that our approach can also be extended to analyze other forms of regularized regression. Our coreset size analysis is based on localizing the function class induced by the regularized objective, specifically by analyzing the sensitivity of the loss function restricted to a region around the regularized solution. This localization reflects the fact that regularization implicitly constrains the functional space, which in turn allows for smaller coresets.
> Although the ridge regression has the optimal coreset [1], for general $\ell_p$ regularized regression introduced in Chhaya et al. (2020) or other variants, the localization framework can, in principle, be adapted to such settings as long as the regularizer imposes sufficient structure to restrict the effective function class. In fact, similar localization techniques have been applied in statistical learning in prior work, such as [2].
>
> [1]. Mai, Tung, et al. Optimal sketching bounds for sparse linear regression. International Conference on Artificial Intelligence and Statistics, 2023.
>
> [2] Pierre C Bellec. Localized gaussian width of m-convex hulls with applications to lasso and convex aggregation. Bernoulli: official journal of the Bernoulli Society for Mathematical Statistics and Probability, 25(4A):3016–3040, 2019.

---

> > ### Comment · Reviewer_FxuS · 2025-11-27
> >
> > I thank the authors for clarifying my doubts. I am satisfied, so I will increase my score to 6.

---

### Official Review · Reviewer_SFut · 2025-11-03

**Soundness:** 3
**Presentation:** 3
**Contribution:** 3
**Rating:** 6
**Confidence:** 1

**Summary:**

The paper considers the problem of coreset construction with sensitivity sampling for LASSO. Note that the coreset construction via sensitivity sampling is a known method. The main contributions of this paper are providing theoretical guarantees for sensitivity sampling in LASSO. The authors then provide some experiments to validate their methods.

**Strengths:**

The paper is clear and easy to follow.

**Weaknesses:**

First things first, I am not an expert in this field, so my evaluation might be unreliable. I did not check the proof, and I will revisit it in the rebuttal phase. Here are my two cents on the paper's weaknesses.

1. Comparison with sketching methods: I see that the experiments of this paper were only conducted with Vanilla LASSO, sensitivity sampling, and uniform sampling. However, they did not compare this method with other approaches for handling big data, like projection methods/sketching. As a general audience, I expect a comparison here, to see if I should really care about sensitivity sampling for LASSO, or if advanced sketching methods for LASSO should work for me/or even work better than the proposed method, when the number of samples ($n$) is large.

2. Comparison with modified LASSO coreset: again, I see that the problems of sensitivity sampling with modified LASSO (using $\|x\|_1^2$ instead of $\|x\|_1$) are also considered. The authors should also compare their method with this one. Although the authors claimed that the modified LASSO introduces some unexpected correlation between features, it is good to know how their performance compares to their proposed approach.

3. Cost of sensitivity score calculation: I see that the computation of the coreset score scales cubically ($O(d^3)$) with the number of features $d$. It should not be the problem for the case $d \ll n$ as the authors considered. However, in a general high-dimensional case, $d$ will scale with $n$, and if $d/n = \alpha$ for a constant $\alpha$, there would be a serious problem. I expect that in such a case, sketching methods would work better. Can the authors comment on this point?

4. Dependence on regularization parameter $\lambda$: I see that the bound on the coreset size includes a term proposional to $1 / \lambda^2$. I expect that things would get ugly when $\lambda$ is close to 0 (e.g., weak regularization). Can the authors comment on this point?

Overall, I am uncertain whether the experiments presented in this paper are sufficient. However, I think that the main punchline of this paper is not totally about the experiments, but the theoretical guarantees the authors established for the coreset construction with sensitivity sampling for LASSO. However, I am not an expert and can only have lukewarm support for this paper, with a low confidence score.

**Questions:**

See above.

---

> ### Author Response · Authors · 2025-11-21
> **Response to Reviewer SFut**
>
> **Question 1**. Comparison with sketching methods: I see that the experiments of this paper were only conducted with Vanilla LASSO, sensitivity sampling, and uniform sampling. However, they did not compare this method with other approaches for handling big data, like projection methods/sketching. As a general audience, I expect a comparison here, to see if I should really care about sensitivity sampling for LASSO, or if advanced sketching methods for LASSO should work for me/or even work better than the proposed method, when the number of samples (n) is large.
>
> Response: We thank the reviewer for pointing out the problem. To address the reviewer’s concern, we have added a new set of experiments comparing our sensitivity sampling-based coreset method with a Gaussian projection-based sketching baseline. This baseline follows the approach of Mai et al. [1], which uses random projections to accelerate LASSO regression.
>
> The new experimental results are included in Appendix C.1 of the revised version. The experimental results show that the proposed LASSO-Sens algorithm generally yields lower regression loss and sparsity than the sketching method, especially on large-scale dataset. On the large-scale mnist8m dataset, LASSO-Sens is up to 10 times faster than the sketching algorithm when the coreset size is set to $m=15\times d$ and $20\times d$. Moreover, on the Synthetic and mnist8m datasets, the sparsity of the LASSO-Sens solution is highly lower than that of the sketching algorithm.
>
> [1]. Mai, Tung, et al. Optimal sketching bounds for sparse linear regression. International Conference on Artificial Intelligence and Statistics, 2023.
>
>
>
> **Question 2**. Comparison with modified LASSO coreset: again, I see that the problems of sensitivity sampling with modified LASSO (using $|x|_1^2$ instead of $|x|_1$) are also considered. The authors should also compare their method with this one. Although the authors claimed that the modified LASSO introduces some unexpected correlation between features, it is good to know how their performance compares to their proposed approach.
>
> Response: Thank you for raising this important question. Our method is designed to solve the standard LASSO objective using the FISTA algorithm, which relies on the proximal operator of the $\ell_1$ norm. This solver is not applicable to the modified LASSO formulation, which involves a squared $\ell_1$ regularization term and lacks an efficient proximal operator. Therefore, directly comparing the two problems with different objective under our framework is unfair.
>
> Chhaya et al. (2020) employed MATLAB’s global optimization toolbox (patternsearch tool) to solve both standard and modified LASSO problems. This solver does not depend on proximal operators and is compatible with both regularization forms, enabling a fair comparison under a unified optimization framework.
>
> We evaluate and compare the LASSO and modified LASSO objective functions under sensitivity sampling on Synthetic dataset, with results reported in the revised version (Appendix C.2). The experimental results show that our method achieves lower regression loss and sparsity across varying values of lambda and coreset size.

---

> ### Author Response · Authors · 2025-11-21
> **Response to Reviewer SFut**
>
> **Question 3**. Cost of sensitivity score calculation: I see that the computation of the coreset score scales cubically (O(d^3)) with the number of features d. It should not be the problem for the case d\ll n as the authors considered. However, in a general high-dimensional case, d will scale with n, and if d/n=\alpha for a constant \alpha, there would be a serious problem. I expect that in such a case, sketching methods would work better. Can the authors comment on this point?
>
> Response. Thank you for the insightful comment. We agree that in high-dimensional settings where $d \gg n$, the $O(d^3)$ cost of computing exact sensitivity scores in our coreset construction may become computationally expensive. However, Chhaya et al. (2020) showed (in Corollary 4.1.1) that the LASSO coreset size cannot smaller than the optimal coreset of $\ell_2$-regression, which requires a coresets of size $\tilde{O}(d/\epsilon^2)$. This implies that in the regime $d \gg n$, the required coreset size exceeds the number of rows $n$, rendering coreset-based methods ineffective in such settings. Therefore, our work focuses on the regime where $n\ll d$,  making coreset both theoretically and practically feasible.
>
> [1]. Chhaya, Rachit, Anirban Dasgupta, and Supratim Shit. On coresets for regularized regression. International Conference on Machine Learning, 2020.
>
>
> **Question 4**. Dependence on regularization parameter $\lambda$: I see that the bound on the coreset size includes a term proposional to $1/\lambda^2$. I expect that things would get ugly when $\lambda$ is close to 0 (e.g., weak regularization). Can the authors comment on this point?
>
> Response. Thank you for the question. The $1/\lambda^2$ term in the coreset size arises from a truncation threshold $\min(1, \frac{log d}{\lambda^2})$. If $\lambda \gg \sqrt{\log d}$, the truncation leads to the coreset size close to $O(d\log(1/\delta))/\epsilon^2)$; otherwise, the truncation becomes ineffective (i.e., the threshold takes value 1), and the coreset size bound becomes $O((d (\log(d))^3)/\epsilon^2+d\log(1/\delta))/\epsilon^2)$.

---

> > ### Comment · Reviewer_SFut · 2025-11-25
> > **Official Comment by reviewer SFut**
> >
> > I thank the authors for their answers. My evaluation remains unchanged.

---

### Author Response · Authors · 2025-11-21
**Response to All Reviewers**

We appreciate the reviewers for their insightful comments and constructive feedback, Below we will address the concerns separately.

To reflect the reviewers’ feedback, we have uploaded an updated version of the manuscript with most of the changes highlighted in blue. The key updates include the following:

- Added a detailed discussion of the relevant literature in the “Other Related Work” section.
- Revised the proof of Theorem 7 in Appendix B.2.
- Included new experiments evaluating the sketching algorithm and the LASSO-Sens algorithm, presented in Appendix C.1.
- Added experimental results comparing sensitivity sampling on the standard LASSO and the modified LASSO objective, detailed in Appendix C.2.

---

### Meta-Review · Area_Chair_BBYa · 2026-01-07

**Summary:**

The submission proposes LASSO-Sens, a sensitivity-sampling coreset construction for the Lasso objective, with sampling probabilities based on LASSO sensitivities. The central idea is to localize the function class induced by the residual + l1 penalty by decomposing it into a residual space and an l1-penalty space, then bounding sampling error via a Gaussian-process reduction + chaining/entropy bounds.

**Reviewer Concerns:**

FxuS asks about a (typo) additive term in Theorem 7 and consistency with Chhaya’s negative result.

VhuG: main concern is experiments (statistical rigor, whether sensitivity beats uniform, and whether runtimes include sensitivity computation). Authors address this and add error bars, clarify 10 trials, and state runtimes include sensitivity computation; reviewer maintains accept.

**Reviewer Scores:**

FxUs mentions increasing their score from 4 -> 6. The rebuttal is substantive: adds sketching comparisons, adds standard vs modified LASSO experiments, adds error bars/10-trial averaging, corrects a typo in the bound and better aligns the discussion with Chhaya et al., and expands related work.

The paper is a clear accept as all revewers are supportive.

---

### Decision · Program_Chairs · 2026-01-26

Accept (Poster)